# Ecological multiplex interactions determine the role of species for parasite spread amplification

**Massimo Stella[1], Sanja Selakovic[2], Alberto Antonioni[3,4,5], Cecilia S Andreazzi[6]***

[1]Institute for Complex Systems Simulation, University of Southampton, Southampton, United Kingdom; [2]Faculty of Geosciences, Utrecht University, Utrecht, Netherlands; [3]Department of Economics, University College London, London, United Kingdom; [4]Grupo Interdisciplinar de Sistemas Complejos, Departamento de Matemáticas, Universidad Carlos III de Madrid, Madrid, Spain; [5]Institute for Biocomputation and Physics of Complex Systems, University of Zaragoza, Zaragoza, Spain; [6]Fiocruz Mata Atlântica, Fundação Oswaldo Cruz, Rio de Janeiro, Brazil

**Abstract** Despite their potential interplay, multiple routes of many disease transmissions are often investigated separately. As a unifying framework for understanding parasite spread through interdependent transmission paths, we present the 'ecomultiplex' model, where the multiple transmission paths among a diverse community of interacting hosts are represented as a spatially explicit multiplex network. We adopt this framework for designing and testing potential control strategies for *Trypanosoma cruzi* spread in two empirical host communities. We show that the ecomultiplex model is an efficient and low data-demanding method to identify which species enhances parasite spread and should thus be a target for control strategies. We also find that the interplay between predator-prey and host-parasite interactions leads to a phenomenon of parasite amplification, in which top predators facilitate *T. cruzi* spread, offering a mechanistic interpretation of previous empirical findings. Our approach can provide novel insights in understanding and controlling parasite spreading in real-world complex systems.
DOI: https://doi.org/10.7554/eLife.32814.001

*For correspondence:
cecilia.andreazzi@fiocruz.br

**Competing interests:** The authors declare that no competing interests exist.

## Introduction

Zoonoses are infections naturally transmitted between animals and humans, and are the most important cause of emerging and re-emerging diseases in humans (*Perkins et al., 2005*; *Jones et al., 2008*; *Lloyd-Smith et al., 2009*). The majority of the zoonotic agents are multi-host pathogens or parasites (*Ostfeld and Holt, 2004*; *Alexander et al., 2012*), whose various host species may differ in their contribution to parasite transmission and persistence over space and time (*Jansen et al., 2015*; *Rushmore et al., 2014*). This heterogeneity of host species contribution to parasite transmission is related to differences in host species' abundance, exposure and susceptibility to infection (*Haydon et al., 2002*; *Altizer et al., 2003*; *Streicker et al., 2013*). Further, many multi-host parasites have complex life cycles with multiple transmission modes, such as vertical, direct contact, sexual, aerosol, vector-borne and/or food-borne (*Webster et al., 2017*).

Among the zoonotic parasites with multiple hosts and transmission modes, *Trypanosoma cruzi* (Kinetoplastida: Trypanosomatidae), a protozoan parasite which causes Chagas disease in humans, has a complex ecology that challenges transmission modelling and disease control (*Noireau et al., 2009*; *Jansen et al., 2015*; *Sosa-Estani and Segura, 2015*). *T. cruzi* has already been found in more than 100 mammalian species and its transmission may be mediated by several interdependent

**eLife digest** Many infectious diseases are contained within a species, so animals from other species are not at risk of catching them. But some diseases – known as zoonoses – can spread between animals and humans. Zoonoses are often transmitted from one host to the next by insects that feed on both animals and humans.

Many well-developed mathematical models exist to understand how infectious diseases are transmitted solely among humans. But modelling how zoonoses spread among all of their hosts is much more difficult. This is because in many cases, the disease can be transmitted in multiple ways – by a contaminated food source or blood-feeding infected insects, or through both wild and domestic animals, complicating the picture further.

To identify what control strategies would be more efficient for reducing the transmission of parasites that can infect multiple host species, Stella et al. created a new mathematical model called the 'ecomultiplex framework'. This model was used to evaluate the complex transmission of Chagas disease, a tropical disease that can be lethal. It combined both ecology (the environment of the Chagas disease parasite) and epidemiology (the characteristics and progress of the disease) to model how the parasites spread among wild animals. By simulating a real-life scenario, Stella et al. were able to identify which host species were most affected, and to test which control strategies would be the most effective in a given environment. The model also revealed that some species may reduce the transmission of the parasite, while others might amplify it, depending on how they interact with other mammals or insects.

The findings will help guide the public-health management of Chagas disease to control transmission more effectively and reduce disease incidence in humans. Besides Chagas disease, many other life-threatening diseases, such as malaria, Leishmaniasis, plague and Lyme disease, are also zoonoses transmitted by multiple ways. The ecomultiplex framework could be of use to ecologists studying these diseases and developing more effective ways to control them.
DOI: https://doi.org/10.7554/eLife.32814.002

mechanisms (*Noireau et al., 2009*; *Jansen et al., 2015*; *Coura et al., 2002*). For instance, *T. cruzi* has a contaminative route of transmission that is mediated by several invertebrate vectors (Triatominae, eng. kissing bug) that gets infected when blood feeding on infected hosts. Susceptible hosts can get infected after scratching and rubbing the parasite-contaminated defecation matter onto the lesion of the bite of an infected vector (*Kribs-Zaleta, 2006*). Transmission may also occur through a trophic route that cascades along the food-web when a susceptible predator feeds on infected vectors or preys (*Noireau et al., 2009*; *Jansen et al., 2015*). In general, sylvatic hosts do not suffer mortality from *T. cruzi* (*Kribs-Zaleta, 2010*) but the parasite establishes a lifelong infection in almost all of them (*Teixeira et al., 2011*).

Chemical insecticides and housing improvement have been the main strategies for controlling Chagas disease in rural and urban areas of Latin America (*Dias and Schofield, 1999*). However, these strategies are proving to be inefficient in reducing transmission (*Roque et al., 2013*). This is possibly related to the maintenance and transmission of parasites among local wild mammalian hosts and its association with sylvatic triatomine vectors (*Roque et al., 2013*; *Roque et al., 2008*). Therefore, modelling parasite transmission in a way that is explicitly considering the ecology of wildlife transmission, is fundamental to understanding and predicting outbreaks.

In this work, we propose to address this challenge through the mathematical framework of multiplex networks (*De Domenico et al., 2013*; *Kivela et al., 2014*; *Boccaletti et al., 2014*; *De Domenico et al., 2016*; *Battiston et al., 2016*), which have been successfully applied to epidemiology (*Lima et al., 2015*; *De Domenico et al., 2016*; *Sanz et al., 2014*) and ecology (*Sonia Kéfi et al., 2015*; *Kéfi et al., 2016*; *Pilosof et al., 2017*; *Stella et al., 2016*). Multiplex networks are multi-layer networks in which multi-relational interactions give rise to a collection of network layers so that the same node can engage in different interactions with different neighbours in each layer (*Kivela et al., 2014*; *Boccaletti et al., 2014*; *De Domenico et al., 2013*).

We study the ecology of multi-host parasite spread by multiple routes of transmission and potential control strategies by developing the 'ecomultiplex' framework (short for ecological multiplex

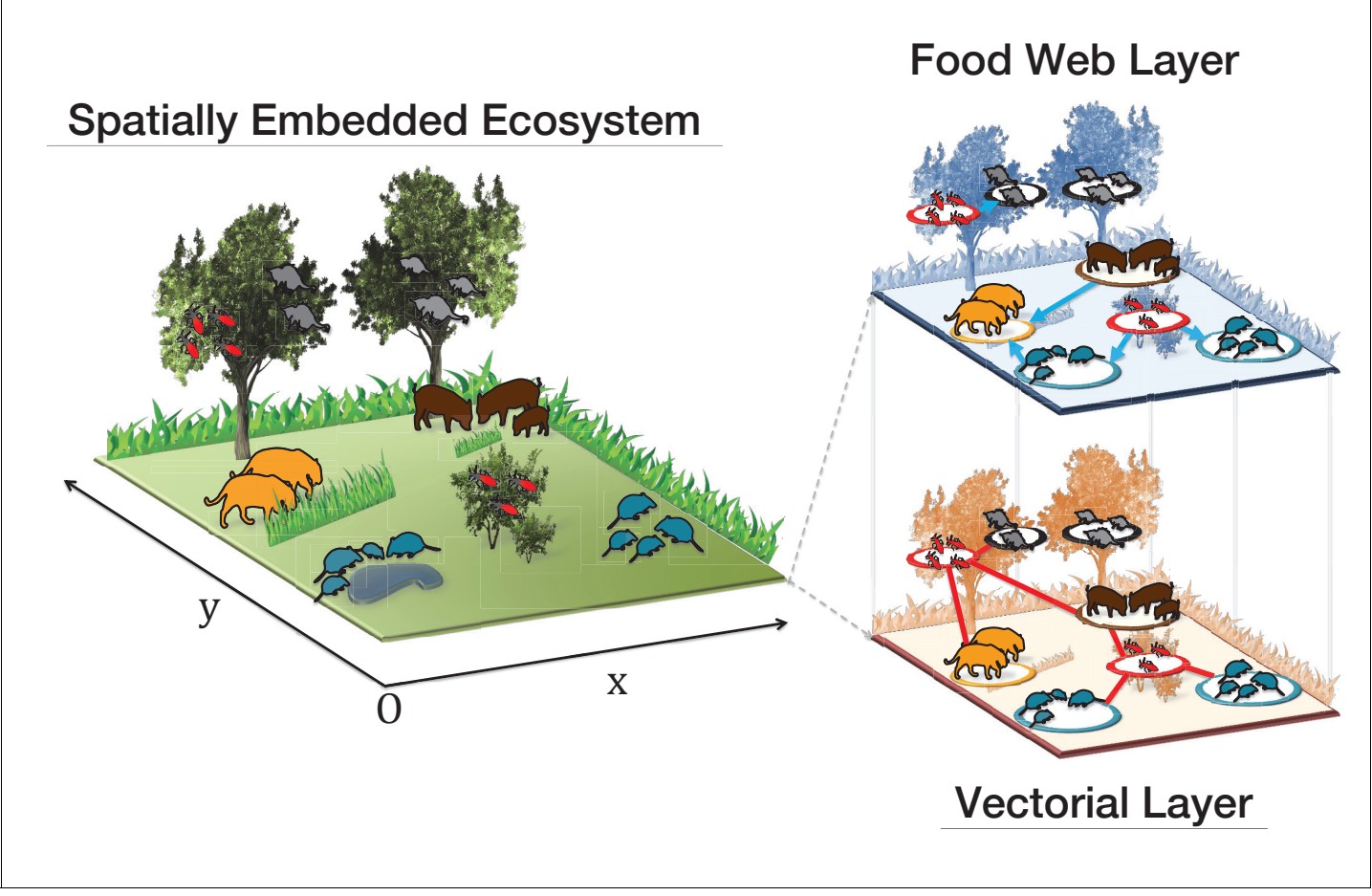

**Figure 1.** Visual representation of our ecological multiplex network model. The background colouring on the right panels indicates that elements such as bushes, trees, grass and water areas are not present in the spatial embedding of the ecomultiplex model, where only the spatial network structure among close animal groups is considered.

DOI: https://doi.org/10.7554/eLife.32814.003

framework), *Figure 1*. This framework is powerful in modelling complex systems such as infectious diseases or parasite transmission in wildlife. Firstly, it allows to account for multiple types of interactions giving rise to parasite transmission with similar or different time scales. Secondly, the ecomultiplex framework uses metabolic theory (*Jetz et al., 2004*) for estimating species frequencies, which are known to influence parasite transmission (*McCallum et al., 2001*). Thirdly, by explicitly considering space, the model also allows to investigate the consequences of spatial structure on parasite transmission (*Hudson et al., 2002*).

The ecomultiplex framework is general in the sense that it can include many ecological interactions among a diverse set of species in a realistic ecosystem. We apply this 'ecomultiplex' formalism to investigating parasite spread in two vector and host communities in Brazil: Canastra (*Rocha et al., 2013*) and Pantanal (*Herrera et al., 2011*). We exploit the theoretical framework enriched with empirical data for designing and comparing different wild host immunisation strategies based on: (i) taxonomic/morphological features (e.g. immunising species belonging to the same taxonomic group or with similar body mass); (ii) species interaction patterns (e.g. immunising species feeding on the vector); and (iii) species' epidemiological role (e.g. immunising species with higher parasite prevalence). Multiplex network structure proved to be an efficient measure in predicting species epidemiological role in both ecosystems. More importantly, considering together multiple transmission mechanisms allowed us to identify a parasite amplification role played by some species of top predators that would not be captured when considering the transmission mechanisms separately.

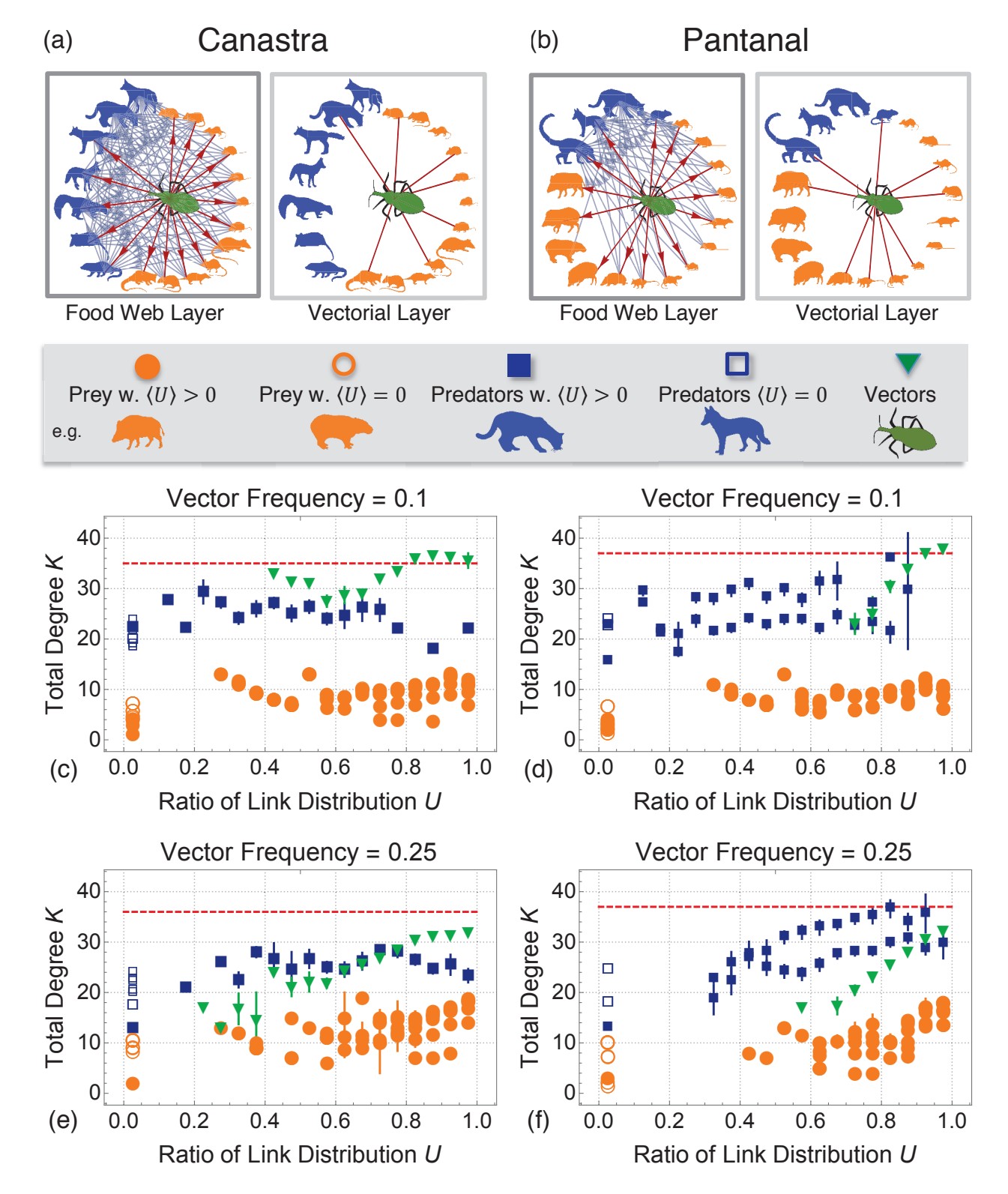

**Figure 2.** Single layer interactions and multiplex cartographies of Canastra and Pantanal biomes. (a) and (b) Food web layer and vectorial layer in Canastra (left) and Pantanal (right) biomes. Predators are highlighted in blue, preys in orange and the vectors in green. Interactions involving the insect are highlighted in red. Interactions involving other species are reported for completeness in blue. (c–f): Multiplex cartography of the Canastra ecomultiplex network with 10% (c) and 25% (e) of total groups as vectors. Multiplex cartography of the Pantanal ecomultiplex network with 10% (d) and

*Figure 2 continued*

25% (f) of total groups as vectors. The red line separates hub nodes, i.e. the most connected nodes within the 95th percentile of the total degree distribution. The cartography highlights the average trends of species: blue for predators, orange for preys, and green for vectors. As evident from (a–e), vectors have higher total degree in the ecosystem and tend to distribute more equally their links across both the multiplex layers than all other species. Vectors are therefore pivotal in the ecosystem.

DOI: https://doi.org/10.7554/eLife.32814.004

## Results

### Network analysis

We start our analysis by investigating the layout of ecological interactions in Canastra and Pantanal (*Figure 2a,b*) obtained from animal diets and parasite infection rates (see also Appendix 1). Multiplex cartography for both Canastra and Pantanal (*Figure 2c,d*) shows that vectors are: (i) more connected and (ii) distribute their links more equally across the ecomultiplex layers than other species. Hence, vectors can get more easily infected in one layer and spread the parasite on another layer with equal likelihood. Hence, the cartography confirms that vectors facilitate parasite spread through their interactions. The local network structure around vectors in Canastra and Pantanal (*Figure 2a,b*) shows that vector colonies are in the Centre of star-like topologies on both layers. These results confirm that kissing bug vector species are pivotal for parasite spread, promoting it on both the food web and the vectorial layer. Although parasite diffusion can be hampered by removing vector colonies from the environment (*Yamagata and Nakagawa, 2006*), these immunisation strategies are not stable as vector reintroduction can happen shortly after elimination (*Funk et al., 2013*). Hence, we focus on immunisation strategies considering vectors' centrality in the ecomultiplex networks but immunising other species.

### Immunisation strategies

As expected, immunising species with the highest parasite infection rate (Hemoculture) is the best strategy for hampering parasite spread for both Canastra and Pantanal (*Figure 3*). This epidemiological strategy slows down parasite spread by almost 30% in Canastra and 26% in Pantanal when the parasite spreads mainly on the food-web layer ($p_v = 0.1$) (*Figure 3*). Immunising species interacting with vectors on the vectorial layer (an ecomultiplex strategy) also performs better than random. The difference between the epidemiological and the ecomultiplex strategies is present only at low vector frequencies ($f_v = 0.1$) in both Canastra (*Figure 3a*) and Pantanal *Figure 3c*) but vanishes when $f_v = 0.25$ and $p_v > 0.2$ (*Figure 3b,d*).

In Canastra, when 10% of the animal groups are vector colonies (*Figure 3a*), biological immunisation strategies are equivalent to immunising species at random. The performance of biological immunisation changes dramatically when vector colonies become more frequent (*Figure 3b*). Immunising large mammals decreases by 12% the global infection time when $p_v = 0.1$, suggesting that large mammals do not facilitate parasite transmission in the model. Immunising all the Didelphidae species leads to similar results (*Figure 3b*). Modest increases in infection time are reported for immunising Cricetidae species when $p_v = 0.2$ (*Figure 3b*). Immunising species feeding on the vector (insectivores) is equivalent to random immunisation (sign Test, p-values>0.1).

In Pantanal, immunising parasitised mammals, parasitised Didelphidae and species with the highest infection rates (Hemoculture 3) are at least two times more effective in slowing down parasite spread compared to other strategies (*Figure 3c,d*). Contrary to what happens in Canastra, when $f_v = 0.1$ and the parasite spreads mainly on the food web ($p \leq 0.2$), immunising parasitised Didelphidae hampers parasite diffusion more than immunising all parasitised mammals (sign Test, p-value<0.01) (*Figure 3c*). Immunising insectivores or large mammals is equivalent to random immunisation (*Figure 3c*). Immunising Cricetidae species always performs worse than random immunisation (*Figure 3c,d*).

### Top predators can lead to parasite amplification

In Canastra, the strategy Hemoculture three includes also immunising one species of top predator, the *Leopardus pardalis* (ocelot) (see *Appendix 7—figure 1*). We compare the performances of Hemoculture three against another immunisation strategy where instead of the ocelot we immunise

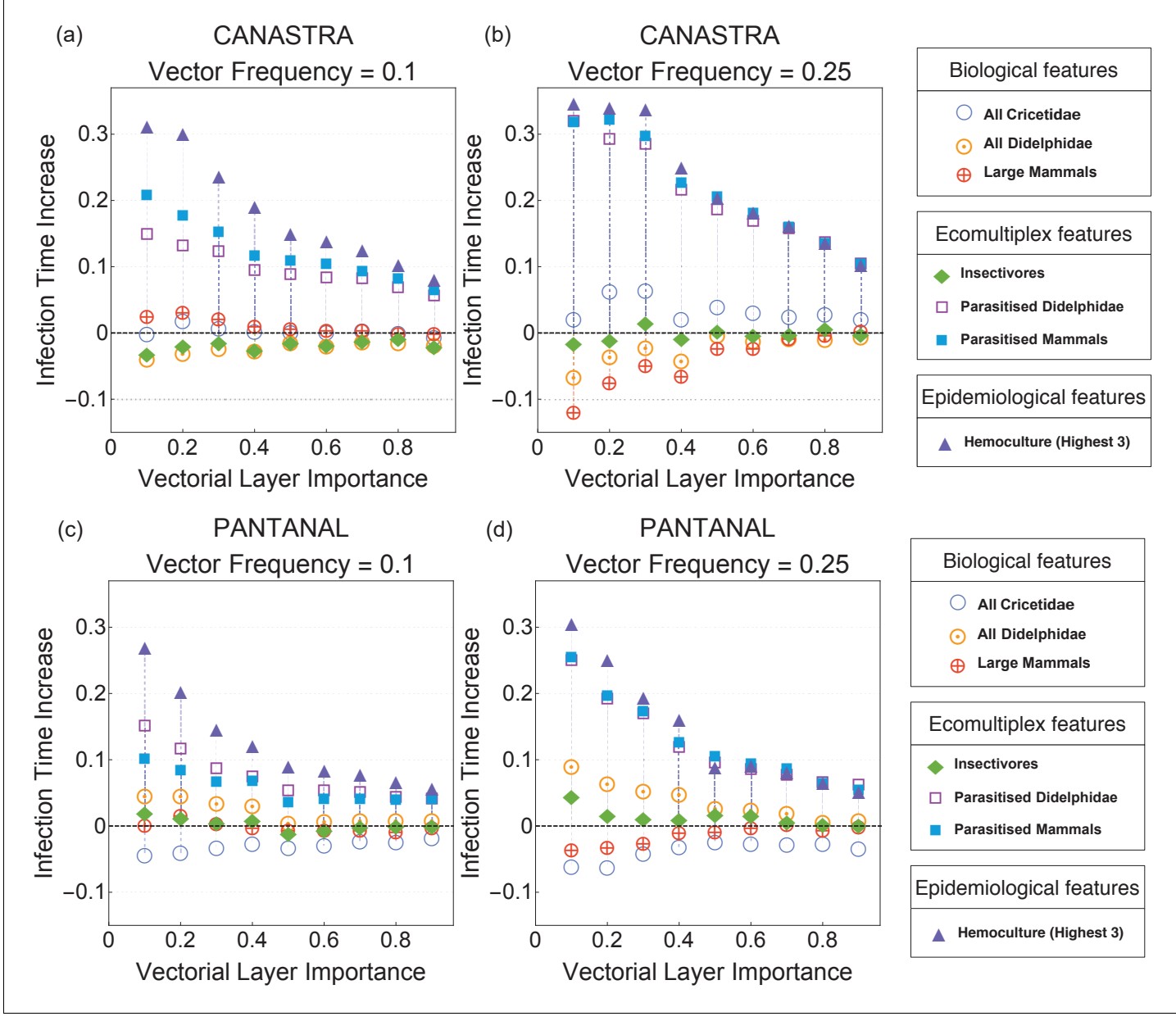

**Figure 3.** Immunisation strategies for the Canastra (top) and Pantanal (bottom) ecosystems when the vector frequency is 0.1 (left) and 0.25 (right). The infection time increase plotted on the y-axis is defined in Immunisation Strategies (Methods and Materials). An increase of +0.3 indicates that the infection time in a given immunisation scenario was 30% higher than in the reference case of random immunisation.

DOI: https://doi.org/10.7554/eLife.32814.005

another top predator, the *Chrysocyon brachyurus* (maned wolf), which had negative *T. cruzi* infection rate in this area (*Rocha et al., 2013*). In general, top predators are related to parasite transmission control in natural environments (*Wobeser, 2013*) so we did not expected differences between different predators.

Instead, results from *Figure 4* indicate a drastic increase of global infection time when a predator with positive parasite infection rate is immunised. This indicates that in Canastra the *Leopardus pardalis* has an amplification effect in spreading the parasite (*Figure 3b*). This phenomenon crucially depends on the ecomultiplex structure, as discussed in the following section.

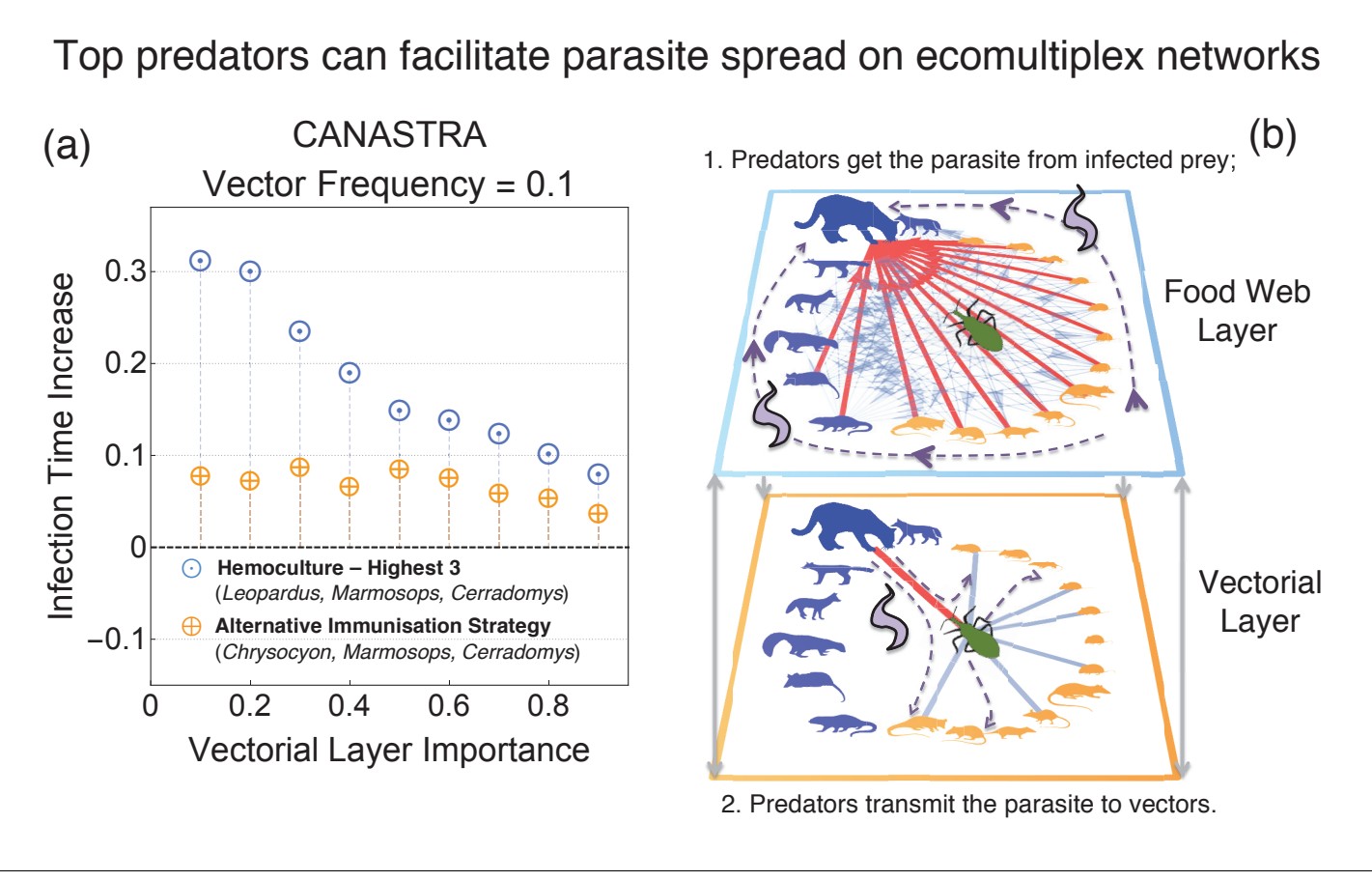

**Figure 4.** Difference in performances of the best immunisation strategy (hemoculture - Highest 3) when another top predator in the ecosystem (not interacting with the vector in the vectorial layer) is immunised instead of *Leopardus pardalis* (hemoculture - H 3 No Leopardus). The other top predator is the maned wolf (*Chrysocyon brachiurus*).

DOI: https://doi.org/10.7554/eLife.32814.006

## Discussion

We introduce ecomultiplex networks as a powerful theoretical framework for modelling transmission of multi-host parasites by multiple routes in species-rich communities. We identify three key points related to the model. Firstly, we show that network structure offers insights on which host species facilitate parasite spread. Secondly, we show that the structure of species interactions can be as useful as epidemiological, taxonomic and morphological traits in controlling parasite transmission. Thirdly, we identify for the first time that the interdependent interactions of top predators affect their functional role in facilitating parasite transmission rather than hampering it.

Ecomultiplex strategies always outperform strategies based on species taxonomic groups, which neglect species' interactions. Further, network structure allows to design immunisation strategies performing as well as strategies considering empirical parasite infection rate, with the advantage of requiring less data. This quantitatively suggests the importance of jointly considering vector-host and predator-prey interactions for understanding *T. cruzi* transmission in wildlife (*Coura, 2006*; *Johnson et al., 2010*; *Penczykowski et al., 2016*). Although Pantanal and Canastra differ in species composition and their ecological interactions, immunising species exposed to contaminative infection through the vectorial layer proves to be efficient at all vectorial layer importances in both ecosystems. This underlines the importance of vectorial transmission for boosting parasite spread also in the food web. Importantly, the ecomultiplex model allows to quantitatively investigate the interplay between the multiple types of interactions leading to parasite transmission, an element conjectured

being crucial for better understanding the ecology of wildlife diseases (*Lafferty et al., 2008*; *Dunne et al., 2013*; *Funk et al., 2013*).

The ecomultiplex model provides insights on how species influence parasite spread. In Pantanal, immunising only Didelphidae with positive parasitaemia (i.e. infection rates) slows down parasite spread more than immunising all mammals with positive parasitaemia. This finding agrees with previous studies that identify Didelphidae as important reservoirs for *T. cruzi* maintenance in natural ecosystems (*Herrera and Urdaneta-Morales, 1992*; *Noireau et al., 2009*; *Coura et al., 2002*). Reservoirs are a system of host populations that are able to maintain the transmission of a given parasite species in space and time (*Haydon et al., 2002*; *Jansen et al., 2015*). Being able to identify the functional role of species based on their topological interactions further highlights the powerfulness of the ecomultiplex framework in modelling parasite diffusion. Notice that the above results and the observed mechanism of parasite amplification are robust also to violations of the assumption of metabolic theory as they are present also in null models with animal abundance independent on body mass (Appendix 8). Model inputs for the ecomultiplex model are partly from published data (e.g. animal diets (*Herrera et al., 2011*) and parasite empirical infection rates (*Herrera et al., 2011*; *Rocha et al., 2013*)). In order to better calibrate and then evaluate model performance, additional observed ecological data should be used. For instance, larger samples for parasite infection rate estimation and, especially, the frequency of animal contacts would both allow for calibration of model parameters such as the SI infection probabilities, which is not feasible with the currently available ecological data.

Within food webs, top predators are generally considered playing a regulating role in parasite spread by preying on infected individuals, thus eliminating infection sources for other animals (*Packer et al., 2003*; *Hatcher et al., 2006*; *Wobeser, 2013*; *Telleria and Tibayrenc, 2017*). Our ecomultiplex framework shows that predators can also facilitate rather than just slow down parasite spread depending on their interactions with vectors. An example is the ocelot in Canastra, which is a generalist predator that feeds on several prey species, including the vectors, and thus has an increased likelihood of becoming infected on the food web (*Figure 4b*). Once infected, ocelots can transmit the parasite to vectors through vectorial interactions. Since vectors facilitate parasite spread, then the ocelot can indeed amplify parasite diffusion. This is true for every generalist predator getting in contact with the vectors. This phenomenon of *parasite amplification*, that is, increased parasite transmission mediated by top predators, emerges only when both ecomultiplex layers are considered together. Therefore, this mechanism remarks the importance of unifying ecological and epidemiological approaches for better modelling parasite transmission. Importantly, this amplification mechanism provides a theoretical explanation by which the ocelot relates with the *T. cruzi* spread, as found in empirical studies (*Rocha et al., 2013*; *Rocha, 2006*). Parasite amplification by predators may also occur in other systems that show multiple transmission routes including trophic transmission, such as in the *Toxoplasma gondii* (*Dubey, 2004*) and *Trypanosoma evansi* (*Herrera et al., 2011*) transmission cycles.

Our theoretical model allows to design and test immunisation strategies in real-world ecosystems by relying on specific assumptions. For instance, since animal groups are embedded in space, home ranges need to be specified for them. For the sake of simplicity, in this ecological version of the model we considered only one effective average interaction radius for all species. Considering species-specific empirical data on spatial distribution represents a challenging yet interesting generalisation for future work when additional ecological data becomes available.

The exposure to parasite infection in the wildlife is mediated by a network of contacts. Consequently, some species are more exposed to the parasite compared to others even when they have the same transmission probability. We consider the same parasite transmission probability across species in the Susceptible-Infected dynamics. By doing that, we give maximum importance to the structure of ecological interactions rather than to the stochasticity of the contagion process. Further, we avoid arbitrary parameter value definition. In our model, the (i) structure of interactions and (ii) the different frequencies of animal groups are analogous to considering different transmission rates. We have already showed that these two elements were sufficient for species to display different probabilities of catching the parasite in our previous work (*Stella et al., 2016*). Immunisation strategies confirm this: immunising species that are more exposed to parasites leads to better immunisation performances compared to random immunisation. Considering species-dependent transmission rates as encapsulated in frequencies and links importantly reduces the number of model parameters.

We assume that parasite spread is happening at much faster rates compared to other meta-population dynamics (e.g extinction or migration), which are not currently considered in the model. However, including meta-population dynamics would allow to explore important research questions such as: (i) the interplay between predation and parasite amplification over top predators influencing parasite spread; (ii) the influence of migration on parasite diffusion; (iii) how extinction patterns influence parasite spread. Implementing the Markovian analytical approach from *Gómez-Gardeñes et al., 2018*) in the ecomultiplex model would allow to reach even more realistic representations of real-world ecosystems.

## Materials and methods

### Ecological multiplex network model

The 'ecomultiplex' model describes an ecological community interacting in a spatially explicit ecosystem, see *Figure 1*, using the novel framework of multiplex networks. Each layer of the ecomultiplex network represents a different route of parasite transmission: (i) food-web interactions and (ii) contaminative interactions mediated by vectors. These infection routes give rise to a multiplex network of two layers where nodes represent groups of individuals of a given species, that is animal groups. Links on the food-web layer are directed to predator species and represent predator-prey interactions. Links on the vectorial layer are undirected and represent insect vector blood meals. Distance among animal groups determines possible interactions: only groups sharing a spatial portion of their home range can interact with each other. We fixed the home range of all animal groups as a circle of radius $r = 0.03$ over a unitary squared space and studied a total of $N = 10000$ animal groups, please see Appendix 3 and[67] for more details regarding the network construction. The small value of $r$ has been tuned in order to keep the ecomultiplex network connected (*De Domenico et al., 2013*) so that the parasite can infect the whole ecomultiplex network.

### Ecological data: trophic interactions and body masses

Different community structures may affect parasite transmission dynamics. We used data from two communities that differed in species composition and interactions (Appendix 1). Predator-prey and vector-host interactions in the ecomultiplex network are based on ecological data related to *T. cruzi* infection in wild hosts within two different areas: Canastra, a tropical savannah in South-Eastern Brazil (*Rocha et al., 2013*) and Pantanal, a vast floodplain in Midwest Brazil (*Herrera et al., 2011*). Both places hold a highly diverse host communities which differ in the structure of interactions, particularly in the vectorial transmission layer (See Appendix 2 for further details).

Trophic interactions in the food web are assigned according to literature data about animals' diets (*AdA et al., 2002*; *VdN, 2007*; *Cavalcanti, 2010*; *de Melo Amboni MP, 2007*; *dos Santos, 2012*; *Reis et al., 2006*; *Rocha, 2006*) (Appendix 1). Since kissing bugs function as a single ecological unit and previous *T. cruzi* epidemiological models treat the vectors as a single compartment

**Table 1.** Immunisation types, names and targets of the strategies we tested (Appendix 1).

| Immunisation type | Strategy name | Strategy targets |
|---|---|---|
| Ecomultiplex Topological Features | Insectivores | Species feeding on the vector in a food-web |
| | Parasitised Didelphidae | Didelphidae contaminated by the vector on a vectorial layer |
| | Parasitised Mammals | All species contaminated by the vector on a vectorial layer |
| Taxonomic/morphological features | All Cricetidae | All Cricetidae |
| | All Didelphidae | All Didelphidae |
| | Large Mammals | All species with a body mass > 1 kg |
| Epidemiological Features | Hemoculture N | The *N* species with the highest likelihood of being found infected with the parasite in field work (see Appendix 1). |
| | Serology N | The *N* species with the highest likelihood of having been infected with the parasite during their life time (see Appendix 1). |

DOI: https://doi.org/10.7554/eLife.32814.007

(*Kribs-Zaleta, 2006*; *Kribs-Zaleta, 2010*), all vector species are grouped as one functional group. Species infection rate is used to estimate the contaminative interactions in the vectorial layer (*Rocha et al., 2013*; *Herrera et al., 2011*). Positive parasitological diagnostics for *T. cruzi* (hemoculture) are used as a proxy for connections on the vectorial layer, since only individuals with positive parasitaemia (i.e. with high parasite loads in their blood) are able to transmit the parasite (*Jansen et al., 2015*). Body masses of host species represent averages over several available references (*Herrera et al., 2011*; *Myers et al., 2008*; *Reis et al., 2006*; *Bonvicino et al., 2008*; *Schofield, 1994*).

## Mathematical formulation for group frequencies

Geographical proximity and biological features regulate link creation in the ecomultiplex model. Species biological features, in particular body masses, regulate the frequency of animal groups (which are all mammals, except for vectors). Previous study *Jetz et al. (2004)* showed that the density $n_i^{-1}$ of individuals of the same species $i$ having the species average body mass $m_i$ follows the metabolic scaling:

$$n_i^{-1} = \beta^{-1} R_i^{-1} m_i^{3/4} \tag{1}$$

where $R_i$ is the species-specific energy supply rate and $\beta$ a constant expressing species metabolism. The above equation comes from metabolic theory and can be used for determining the scaling relationship between body mass $m_i$ and frequency $f_i$ of animal groups (rather than animal individual) for species $i$, depending on the frequency of vector colonies $f_v$ (Appendix 3):

$$f_i = (1 - f_v) \frac{m_i^{-1/4}}{\sum_{j=1} m_j^{-1/4}}. \tag{2}$$

As a consequence of metabolic theory, the frequencies of animal groups in our ecomultiplex model scale as a power-law of body mass with exponent $-1/4$ rather than $-3/4$ (which is the scaling exponent for individuals rather than groups). We explicitly leave $f_v$ as a free parameter of the model in order to investigate the influence of the frequency of vector colonies on parasite spreading.

## Metrics for multiplex network analysis

We investigate the structure of a given ecomultiplex network through the concept of multiplex cartography (*Battiston et al., 2014*), see also Appendix 5 for the definition of multiplex cartography. In our case, the cartography describes how individual groups engage into trophic interactions on the ecomultiplex structure by considering: (i) the total number K of trophic interactions an animal group is involved in and (ii) the ratio U of uniform link distribution across layers, which ranges between 0 (all the links of a group are focused in one layer) and 1 (all links of a node are uniformly distributed across layers). The higher K, the more an animal group interacts with other groups. The higher U the more an animal group will engage in feeding and vectorial interactions with the same frequency. The multiplex cartography for Canastra and Pantanal is reported and discussed in Appendix 5.

## Susceptible-Infected model on the ecological multiplex network

As explained in the introduction, we focus on parasites causing lifelong infections in wild hosts. Hence, parasite spread is simulated as a Susceptible-Infected (SI) process (*Hastings and Gross, 2012*): Animal groups are susceptible or infectious. We assume that parasite transmission among animal groups happens considerably faster than both (i) group creation or extinction and (ii) parasite transmission within groups, so that fixed numbers of hosts and vectors can be considered, as in previous works (*Keeling and Rohani, 2008*; *Legros and Bonhoeffer, 2016*). At each time step, the parasite can spread from an infected group to another one along a connection either in the vectorial (with probability $p_v$) or food-web (with probability $1 - p_v$) layer. We consider $p_v$ as a free parameter called vectorial layer importance, that is the rate at which transmission occurs through the consumption of blood by vectors rather than predator-prey interactions. We assume that all the species have the same probability of getting infected, since the group gets infected if it interacts with an infected group. However, the transmission rate is the outcome of the probability of infection, species groups frequency in the environment and the interactions in the ecomultiplex. We characterise globally the

SI dynamics by defining the *infection time* $t^*$ as the minimum time necessary for the parasite to reach its maximum spread within the networked ecosystem (*Stella et al., 2016*). The infection starts from a small circle of radius 0.03 in the middle of the unitary space infecting all animal groups within that area. Initial conditions are randomised over different simulations.

## Immunisation strategies

Immunisation strategies provide information on how species influence parasite spread: immunising species that facilitate parasite spreading should increase the global infection time $t^*$ compared to immunising random species. We focus on immunising only $10\%$ of animal groups in ecomultiplex networks with 10000 nodes, in either high ($f_v = 0.25$) or low vector frequency scenarios ($f_v = 0.1$). By immunising groups at random in ecomultiplex networks with $N$= 10000 nodes, we identify φ = 1000 as the minimum number of groups/nodes that have to be immunised in order to observe increases in $t^*$ compared to random immunisation with a significance level of 5%. Immunised groups are selected according to three categories of immunisation strategies (*Table 1*):

- Taxonomic/morphological features: main taxonomic groups or body mass;
- Ecomultiplex network features: interaction patterns on the ecomultiplex structure;
- Epidemiological features: Hemoculture and serological diagnostic measures of parasite infection rate in wildlife.

We define the infection time increase $\Delta t_s$ as the normalised difference between the median infection time $t_s$ when $\phi = 1000$ nodes are immunised according to the strategy $s$ and the median infection time $t_r$ when the same number of nodes is immunised uniformly at random among all animal groups, $\Delta t_s = \frac{t_s - t_r}{t_r}$. Infection times are averages sampled from 500 simulated replicates. Differences are always tested at 95% confidence level.

Positive increases imply that the immunisation strategy slowed down the parasite in reaching its maximum spread over the whole ecosystem more than random immunisation. Negative increases imply that random immunisation performs better than the given immunisation strategy in hampering parasite diffusion.

## Model inputs, parameters used and model outputs

Summing up, the ecomultiplex model adopts the following parameters (Appendix 4):

- Number of total animal groups $N$. We set $N = 10000$ for numerically robust results but the same results were observed also at $N = 1000$ and $N = 500$.
- Frequency of vector colonies $f_v$. We explore low ($f_v = 0.1$) and high ($f_v = 0.25$) scenarios of vector frequency;
- Interaction radius $r$: any two animal groups are connected only if they are closer than $r$ in space. $r$ was tuned numerically to $r = 0.03$ for getting connected multiplex networks (Appendix 4);
- Probability $\beta$ for an infected node of transmitting the infection to a susceptible node in the SI model (transmission rate). We chose $\beta = 1$ in order to be compatible with previous results (*Stella et al., 2016*) (Appendix 4);
- Vectorial layer importance $p_v$, determining the likelihood with which the parasite spreads along a link in the vectorial layer rather than using a link in the food web;
- Number of immunised animal groups $\Phi$. We numerically set $\Phi = 1000$ for obtaining statistically significant increases in the global infection time $t^*$ compared to random immunisation within a significance level of 0.05.

The ecomultiplex model also considers the following ecological data as inputs:

- Average body mass $m_i$ for individuals of species $i$. These represent inputs from ecological data (*Herrera et al., 2011*; *Rocha et al., 2013*) and are used for computing frequencies of animal groups (Appendix 3);
- Ecological predator-prey and vectorial interactions, respectively determined from animal diets and parasite infection rates (Appendix 1).

As outputs the model produces the dynamics of parasite spreading. The total number $N_{inf}$ of infected animal groups was found to be constant across different immunisation strategies, vectorial probabilities $p_v$ and ecosystems, $N_{inf} = 8700 \pm 100$ or $N_{inf} \approx (97 \pm 1)\%$ of susceptible hosts, in terms of

model outputs we focus on the time necessary for the parasite to reach its maximum spread, that is, on the global infection time.

## Acknowledgements

The authors thank Alireza Goudarzi, Paula Lemos-Costa, Renske Vroomans, Enrico Sandro Colizzi, Ayana Martins, Markus Brede, Manlio De Domenico and Cole Mathis for insightful discussions and acknowledge the WWCS2017.

## Additional information

### Funding

| Funder | Grant reference number | Author |
|---|---|---|
| Schweizerischer Nationalfonds zur Förderung der Wissenschaftlichen Forschung | P2LAP1-161864 | Alberto Antonioni |
| Netherlands Organization for Scientific Research | 645.000.013 | Sanja Selakovic |
| Engineer Research and Development Center | EP/G03690X/1 | Massimo Stella |
| Netherlands Organization for Scientific Research | 647570 | Sanja Selakovic |
| Schweizerischer Nationalfonds zur Förderung der Wissenschaftlichen Forschung | P300P1-171537 | Alberto Antonioni |

The funders had no role in study design, data collection and interpretation, or the decision to submit the work for publication.

### Author contributions

Massimo Stella, Conceptualization, Methodology, Formal analysis, Writing—original draft, Writing—review and editing; Sanja Selakovic, Alberto Antonioni, Conceptualization, Methodology, Writing—original draft; Cecilia S Andreazzi, Conceptualization, Investigation, Methodology, Writing—original draft, Writing—review and editing

### Author ORCIDs

Massimo Stella https://orcid.org/0000-0003-1810-9699
Alberto Antonioni http://orcid.org/0000-0002-5788-3348
Cecilia S Andreazzi http://orcid.org/0000-0002-9817-0635

### Decision letter and Author response

Decision letter https://doi.org/10.7554/eLife.32814.028
Author response https://doi.org/10.7554/eLife.32814.029

## Additional files

### Supplementary files

• Transparent reporting form
DOI: https://doi.org/10.7554/eLife.32814.008

### Data availability

All data generated or analysed during this study are included in the manuscript and appendices.

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

# Appendix 1

DOI: https://doi.org/10.7554/eLife.32814.009

## Dataset

We parametrised the ecomultiplex models with data on species ecology (*Appendix 1—table 1* and *2*) and interactions (*Appendix 1—figure 1* and *2*) in two localities: Canastra (*Rocha et al., 2013*) and Pantanal (*Herrera et al., 2011*). We choose these two datasets because they are the most complete studies describing host species diversity and infection prevalence of *T. cruzi* in natural environments for a long period. These studies were conducted for a long period of time with relatively high sampling effort, compared to other surveys of *T. cruzi* in the wild.

We obtained species body masses from the literature (*Herrera et al., 2011*; *Myers et al., 2008*; *Reis et al., 2006*; *Bonvicino et al., 2008*; *Schofield, 1994*) and calculated the average body mass when more than one reference was available for the same species. Interaction matrices contain data about existing trophic and contaminative interactions. The former are represented as blue squares while the latter as green ones. Predators are represented in columns and prey in rows. Trophic interactions in the food weblayer were assigned according to literature information about animals' diets and ecology in Brazil (*AdA et al., 2002*; *VdN, 2007*; *Cavalcanti, 2010*; *de Melo Amboni MP, 2007*; *dos Santos, 2012*; *Reis et al., 2006*; *Rocha, 2006*).

These studies classify prey species into two main categories: "invertebrates" and ßmall mammals". We assigned a potential trophic interaction between a predator and a prey if the prey species belong to the prey category reported on species' diet For example, we assign a potential trophic interaction between Cerdocyon thous and Akodon spp because Cerdocyon thous is reported to have preyed on small mammals. However, we constrained these interactions based on information about species ecology such as use of habitat. For instance if a potential small mammal prey species is arboreal (e.g. Caluromys philander) we only considered predator species that are also able to use the arboreal strata (e.g. Leopardus pardalis). The study conducted in the Canastra area (herein Canastra ecomultiplex network) was located within the Serra da Canastra National Park and adjacent areas, in Minas Gerais state, South Eastern Brazil. It is an important remnant of the Cerrado biome, which is a vast tropical savana (*Rocha et al., 2013*). The Pantanal study (Pantanal ecomultiplex network) was conducted in the southern Mato Grosso do Sul state, mildwest Brazil. The Pantanal biome is a vast floodplain formed by a mosaic of seasonally inundated native grasslands, savannas, scrub and semi-deciduous forests.

Positive parasitological diagnostics for *T. cruzi* (Hemoculture) (*Rocha et al., 2013*; *Herrera et al., 2011*) were used as a proxy for the interactions in the vectorial layer, since only individuals with positive parasitaemia (i.e. with high parasite loads in their blood) are able to transmit the parasite to vectors (*Jansen et al., 2015*). Species interactions differed between places because the species rate of infection also differed. For example, Leopardus pardalis interacts with the vector in the vectorial transmission layer in Canastra, but not in Pantanal because of different parasite infection rate data.

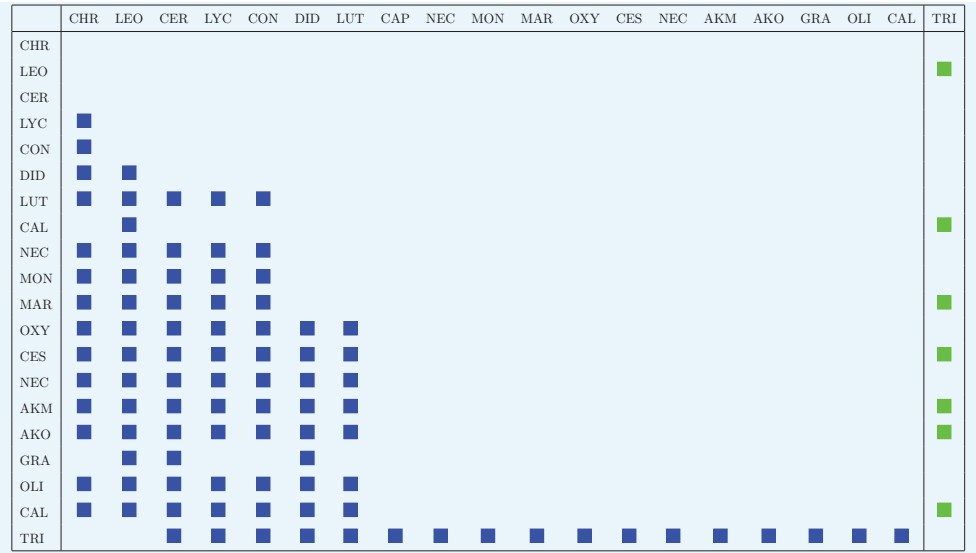

**Appendix 1—figure 1.** Interaction matrix includes trophic (predator-prey, blue squares) and vectorial (vector-host, green squares) interactions in Canastra area.
DOI: https://doi.org/10.7554/eLife.32814.010

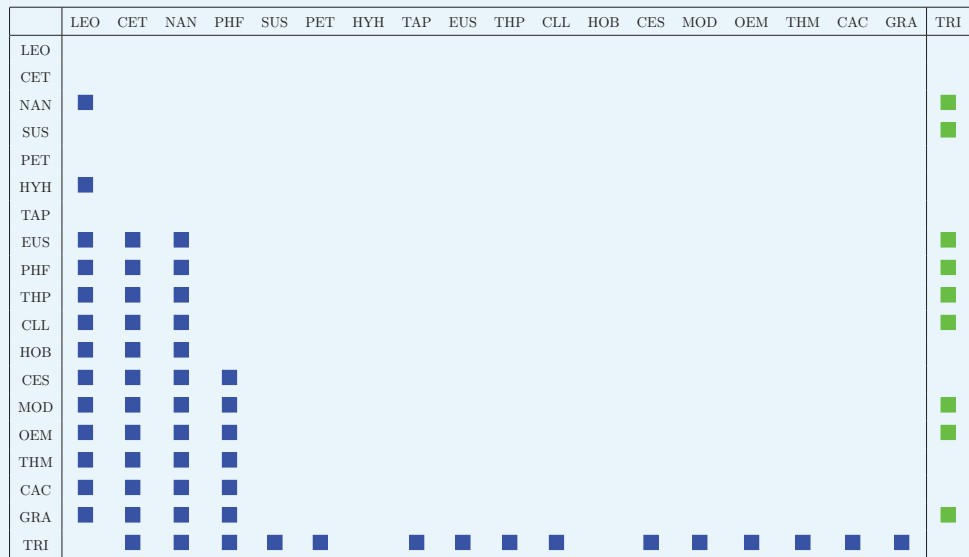

**Appendix 1—figure 2.** Interaction matrix includes trophic (predator-prey, blue squares) and vectorial (vector-host, green squares) interactions in Pantanal area.
DOI: https://doi.org/10.7554/eLife.32814.011

**Appendix 1—table 1.** Taxonomic and ecological data of different animal species in Canastra area (1- (*Reis et al., 2006*), 2- (*Myers et al., 2008*), 3- (*Herrera et al., 2011*), 4- (*Bonvicino et al., 2008*), 5- (*Schofield, 1994*)). (*Rocha et al., 2013*) report prevalence measurements for some genera, indicating the species that were collected during the study. Because those species are ecologically similar we calculated the average body size of the genus considering species that were collected. [a]M. americana, M. domestica and M. sorex. [b]A. sp., A. lindberghi, A. cursor. [c]O. sp., O. nigripes, O. rupestris. [d]C. sp, C. tener.

| ID | Species | Common name | Family | Diet type | Biomass [gr] | References |
|---|---|---|---|---|---|---|
| CHR | *Chrysocyon brachyurus* | Maned wolf | Canidae | omnivorous | 25000 | [1, 2] |
| LEO | *Leopardus pardalis* | Ocelot | Felidae | carnivorous | 9740 | [1, 2, 3] |
| CER | *Cerdocyon thous* | Crab-eating fox | Canidae | omnivorous | 6600 | [1, 2, 3] |
| LYC | *Lycalopex vetulus* | Hoary fox | Canidae | omnivorous | 3350 | [1, 2] |
| CON | *Conepatus semistriatus* | Stripedhog-nosed skunk | Mustelidae | omnivorous | 2567 | [1, 2] |
| DID | *Didelphis albiventris* | White-eared opossum | Didelphidae | omnivorous | 1625 | [1, 2] |
| LUT | *Lutreolina crassicaudata* | Thick-tailed opossum | Didelphidae | omnivorous | 600 | [1, 2] |
| CAP | *Caluromys philander* | Bare-tailedwoolly opossum | Didelphidae | omnivorous | 255 | [1, 2] |
| NEC | *Nectomys squamipes* | South American water rat | Cricetidae | omnivorous | 270 | [1, 2] |
| MON | *Monodelphis spp a* | Short-tailed opossum | Didelphidae | omnivorous | 72.8 | [1, 2, 3] |
| MAR | *Marmosops incanus* | Gray slander opossum | Didelphidae | omnivorous | 80 | [1, 2] |
| OXY | *Oxymycterus delator* | Spy hocicudo | Cricetidae | omnivorous | 78.3 | [4] |
| CES | *Cerradomys subflavus* | Rice rat | Cricetidae | omnivorous | 73 | [4] |
| NEC | *Necromys lasiurus* | Hairy-tailed bolo mouse | Cricetidae | omnivorous | 52 | [1, 2, 4] |
| AKM | *Akodon montensis* | Montane grass mouse | Cricetidae | omnivorous | 40.5 | [1, 2, 4] |
| AKO | *Akodon spp b* | Grass mouse | Cricetidae | omnivorous | 33.25 | [1, 4] |
| GRA | *Gracilinanus agilis* | Agile gracile opossum | Didelphidae | omnivorous | 27.25 | [1, 2, 3] |
| OLI | *Oligoryzomys spp c* | Pygmy rice rats | Cricetidae | omnivorous | 21.23 | [1, 4] |
| CAL | *Calomys spp d* | Vesper mouse | Cricetidae | herbivorous | 18.65 | [1, 4] |
| TRI | *Triatominae* | Kissing bug | Reduviidae | blood | 0.2 | [5] |

DOI: https://doi.org/10.7554/eLife.32814.012

**Appendix 1—table 2.** Taxonomic and ecological data of different animal species in Pantanal area (1- (*Reis et al., 2006*), 2- (*Myers et al., 2008*), 3- (*Herrera et al., 2011*), 4- (*Bonvicino et al., 2008*), 5- (*Schofield, 1994*)).

| ID | Species | Common name | Family | Diet type | Biomass [gr] | References |
|---|---|---|---|---|---|---|
| LEO | *Leopardus pardalis* | Ocelot | Felidae | carnivorous | 9740 | [1, 2, 3] |
| CET | *Cerdocyon thous* | Crab-eating fox | Canidae | omnivorous | 6600 | [1, 2, 3] |
| NAN | *Nasua nasua* | South American coati | Procyonidae | omnivorous | 5140 | [1, 2, 3] |
| SUS | *Sus scrofa* | Wild boar | Suidae | omnivorous | 105750 | [1, 2, 3] |
| PET | *Pecari tajacu* | Collared peccary | Tayassuidae | omnivorous | 24000 | [1, 2, 3] |
| HYH | *Hydrochaeris hydrochaeris* | Capybara | Caviidae | herbivorous | 46200 | [1, 2, 3] |
| TAP | *Tayassu pecari* | White-lipped peccary | Tayassuidae | omnivorous | 30200 | [1, 2, 3] |
| EUS | *Euphractus sexcinctus* | Six-banded armadillo | Chlamyphoridae | omnivorous | 4450 | [1, 2, 3] |
| PHF | *Philander frenatus* | Southeastern four-eyed opossum | Didelphidae | omnivorous | 306.2 | [1, 2, 3] |
| THP | *Thrichomys pachyurus* | Paraguayan punar | Echimyidae | herbivorous | 291.25 | [1, 3, 4] |
| CLL | *Clyomys laticeps* | Broad-headed spiny rat | Cricetidae | herbivorous | 187.33 | [1, 3, 4] |
| HOB | *Holochilus brasiliensis* | Web-footed marsh rat | Cricetidae | herbivorous | 196 | [1, 3, 4] |
| CES | *Cerradomys scotti* | Lindbergh's rice rat | Cricetidae | omnivorous | 92.34 | [3, 4] |
| MOD | *Monodelphis domestica* | Gray short-tailed opossum | Didelphidae | omnivorous | 111.2 | [1, 2, 3] |
| OEM | *Oecomys mamorae* | Mamore arboreal rice rat | Cricetidae | herbivorous | 83.75 | [1, 3, 4] |
| THM | *Thylamys macrurus* | Long-tailed fat-tailed opossum | Didelphidae | omnivorous | 46.6 | [1, 2, 3] |
| CAC | *Calomys callosus* | Large vesper mouse | Cricetidae | herbivorous | 37.45 | [1, 3] |
| GRA | *Gracilinanus agilis* | Agile gracile opposum | Didelphidae | omnivorous | 27.25 | [1, 2, 3] |
| TRI | *Triatominae* | Kissing bug | Reduviidae | blood | 0.2 | [5] |

DOI: https://doi.org/10.7554/eLife.32814.013

## Appendix 2

DOI: https://doi.org/10.7554/eLife.32814.014

### Ecology of *T. cruzi* vectors

Approximately 140 species of triatomine bugs have been identified worldwide, however only a few are competent vectors for *T. cruzi* (***Lent and Wygodzinsky, 1979***; ***Schofield et al., 2009***; ***Rassi Jr et al., 2010***). Of the 16 triatomine species found in Amazonian Brazil, around ten are infected with *T. cruzi* (***Coura et al., 2002***). The most important vectors belong to *Triatoma*, *Rhodnius* and *Panstrongylus* genera. The vector distribution in Brazil is wide (***Browne et al., 2017***), and ecologic niche modeling results show that all environmental conditions in the country are suitable to one or more of the potential vectors that transmit *T. cruzi* (***Gurgel-Gonçalves et al., 2012***). The model shows two species that would be commonly found in both Canastra and Pantanal: (*T. pseudomaculata* and *T. sordida*), while four species are unique to Canastra: (*T. melanica, T. tibiamaculata, T. vitticeps* and *T. wygodzinsky* three are unique to Pantanal: (*T. matogrossensis, T. vandae* and *T. williami*). Although the diversity of vector species partly differs among the two localities, most of them have similar habitat preferences. Vectors usually prefer to inhabit birds nests on palm trees, trees barks and mammals' burrows (***Gurgel-Gonçalves et al., 2012***; ***Abad-Franch et al., 2009***).

Analogously to previous modelling approaches to parasite spreading (***Kribs-Zaleta, 2006***; ***Kribs-Zaleta, 2010***), in our model different species of vectors are represented as one functional compartment because of their similar biology.

# Appendix 3

DOI: https://doi.org/10.7554/eLife.32814.015

## Mathematical formulation of scaling laws for the frequency of animal groups

In this section we report an extended version of the section 'Mathematical formulation for group frequencies' providing additional details on the derivation of the animal group frequencies we used for our model.

Spatial embedding and ecological data regulate link creation by determining which species engage in predator-prey or parasite-host interactions in our ecomultiplex model. Ecological data, in particular body masses, also determine the expected species biomass and density in the environment using metabolic theory (**Robinson and Redford, 1986**; **Jetz et al., 2004**). In other words, the ecomultiplex model assumes that metabolic theory provides a good approximation for the frequency of animal groups of a given species in the ecosystem.

Let us now derive the frequencies $f_i$ of animal groups of a given species $i$ from metabolic theory. Consider an ecosystem with $N$ animal groups, $S$ mammal species and one species of vector insect. We denote the number of animal groups of species $i$ as $N_i$, such that the frequency of groups of animals of that species is defined as:

$$f_i = \frac{N_i}{N}. \tag{3}$$

The $N_i$s for different species are subject to a constraint: they have to sum up to the total number of animal groups $N$ in the ecosystem minus the number $N_v$ of vector colonies, $\sum_{i=1}^{S} N_i = N - N_v$. If we represent the frequency of vector colonies as $f_v = \frac{N_v}{N}$, we can divide the previous sum by $N$ such that,

$$\sum_{i=1}^{S} \frac{N_i}{N} = 1 - \frac{N_v}{N} \rightarrow \sum_{i=1}^{S} f_i = 1 - f_v. \tag{4}$$

In other words, all the frequencies of animal groups and the frequency of vector colonies have to sum up to 1. Notice that $f_i$ is the fraction of groups of species $i$ in the ecosystem and in order to compute it from metabolic theory, an expression for the number of groups $N_i$ as a function of the body mass $m_i$ has to be derived. We assume that animal groups of the same species $i \in 1, ..., S$ contain an average number of individuals $n_i$ with average body mass $m_i$. Therefore, within an animal group there will be a total body mass equal to $m_i$ times the average number of animals composing the group $n_i$. We define the total average body mass $M_i$ of animals from species $i$ in the ecosystem as the total body mass of animals in an animal group $(m_i n_i)$ times the number of animal groups $N_i$ of species $i$, in formulas:

$$M_i = (m_i n_i) N_i = m_i n_i f_i N, \tag{5}$$

where $n_i$ represents the number of individuals within a given animal group exploiting a given home range. Previous literature (**Jetz et al., 2004**) show that the minimum density $n_i^{-1}$ of individuals within a home range that is sufficient to sustain their metabolic requirements that scales as:

$$n_i^{-1} = \beta^{-1} R_i^{-1} m_i^{3/4} \tag{6}$$

where $R_i$ is the species-specific energy supply rate, that is, the energy resources necessary to sustain the animal group in a given area and unit of time, expressed in $W/km^2$. $\beta$ is a normalisation constant related to the species metabolism. Empirical work has shown that $R_i$ is roughly independent on body mass (**Jetz et al., 2004**). Inserting **Equation 1** in **Equation 5**, we obtain:

$$M_i = m_i n_i f_i N = m_i \left( \beta R_i m_i^{-3/4} \right) f_i N. \tag{7}$$

*Equation 7* relies on two assumptions: (i) metabolic theory provides a good approximation for species densities and (ii) there is an average size $n_i$ across animal groups of a given species. *Equation 7* can be further used for determining how the frequency of animal groups $f_i$ scales with body mass. To the best of our knowledge, there is no empirical data relating the frequency of animal groups with total body masses in ecosystems. However it is known from metabolic theory that the mass specific expenditure $c$ of energy $E_i$ per lifespan $\Delta T_i$ of an individual of mass $m_i$ of species $i$ is independent on body mass and it does not differ across different species (*Speakman, 2005*).

The mass specific expenditure rate $c$ can be written as:

$$c = \frac{E_i}{\Delta T_i m_i} \propto m_i^0. \tag{8}$$

Dividing the expenditure $c$ by the home range area $H_i$ over which energy $E_i$ is gathered by an individual animal of species $i$, then we have that:

$$\frac{c}{H_i} = \frac{E_i}{\Delta T_i H_i m_i} = \frac{R_i}{m_i}, \tag{9}$$

where $R_i = E_i/(\Delta T_i H_i)$ follows its definition (i.e. the energy available in the home range per unit of area and time). *Equation 9* is compatible with the findings from *Jetz et al. (2004)*, where $R_i \propto m_i^o$ and $H_i \propto m_i$ across species. Under the assumption that metabolic theory is a valid approximation regulating the density of individuals in space according to their body masses, we found that $\frac{c}{H_i} = \frac{R_i}{m_i}$ or rather that:

$$m_i = \frac{H_i}{c} R_i \tag{10}$$

We want to use *Equation 10* as an additional constraint for solving *Equation 7* for the frequencies of animal groups. In *Equation 7* we have total body masses $M_i$, therefore we consider summing all the body masses of the individual animals of species $i$ corresponding to average body mass $m_i$. Since each of the $N_i$ animal groups will contain $n_i$ individuals, then the biomass will be:

$$M_i = N_i n_i m_i = \sum_{l=1}^{N_i n_i} m_i = \sum_{l=1}^{N_i n_i} \frac{H_i}{c} R_i \tag{11}$$

However *Equation 11* over-estimates the total body mass because home ranges of animal groups are not independent from each other (i.e. disjoint) but they rather overlap. Overlap in home ranges might lead to competition and thus species groups might get less resources than the energy provided by the whole home range $H_i$. To see this, consider the simple example in which two animal groups of the same species have an overlap $\Omega$ in their home ranges (*Appendix 3—figure 1*, left). Each of the animal groups will not exploit their own home range $H_i$ because the energy coming from $\Omega$ will have to be shared with the other animal group of the same species. Each animal group will then use an 'effective' home range equal to $H_i - O/2$, where the negative term considers the energy sharing with the other group. In the more complex case where many animal groups can compete with each other, not all the home range will contribute to the energy transformed into body mass, so that it is not expected that $M_i = sum_{h=1}^{N_i n_i} \frac{H_i}{c} R_i$. Since the effect of this overlap is poorly constrained by empirical data, we approximate the energy effectively available to animal groups by considering 'effective' home ranges $H_i^*$ that on averaged not overlap with each other, i.e. with radius equal to half the average distance $d_i$ between animal groups of species $i$ (*Appendix 3 Figure 1*, right). Because of its definition, on average the radius of an effective home range will be $r_i^* = d_i/2 = \sqrt{H_i^*/Pi}$. As a consequence of reducing overlap, the effective home range for species $i$ will be smaller

than the usual home range, thus keeping into account the reduced energy availability to animal groups due to overlap and competition for resources in the same ecosystem. In mathematical terms, considering effective home ranges allows to compute the total biomass for species $i$ as the sum of independent (non-overlapping) elements $\frac{H_i^*}{c} R_i$.

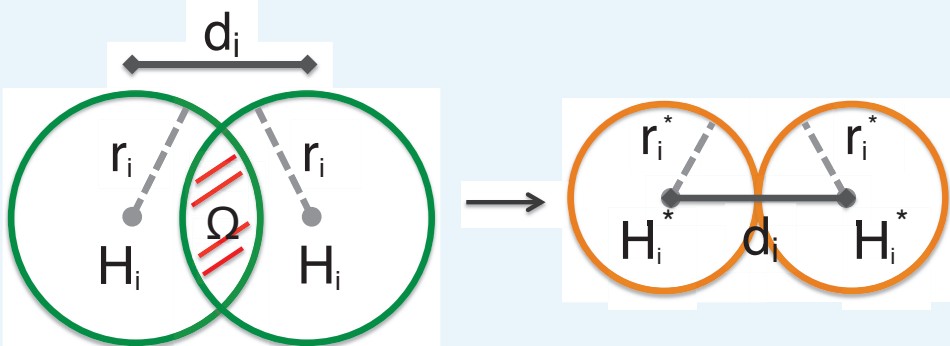

**Appendix 3—figure 1.** In general home ranges $H_i$ for different animal groups of the same species can overlap (e.g. see the red overlapping area $\Omega$ on the left). Hence, not all the energy available in the home range can sustain the biomass of the animal groups as some resources must be shared and competition can occur. We therefore approximate the total energy effectively available from the overlapping home ranges with the energy coming from non-overlapping effective home ranges $H_i^*$, which allow to approximate the total biomass for species $i$. Effective home ranges $H_i^*$ have radius $r^*$ equal to half the distance between animal groups. .
DOI: https://doi.org/10.7554/eLife.32814.016

Furthermore, because we assume a uniform spatial embedding of animal groups over space and a large network representation, the average distance between animal groups $d_i$ is the same across species so that $r_i^* = r^*$. Notice that while it is biologically known that home ranges $H_i$ can greatly vary across animal species (**Jetz et al., 2004**), the effective home range $H^*$ is rather a function of the average distance of species, which depends on the spatial embedding of species. Without ecological data considering how animal groups are embedded in real-world ecosystems, a uniform random embedding is the simplest assumption possible, which is adopted in our ecomultiplex model. These assumptions support our adoption of an 'effective' home range $H^*$ for all animal groups, so that for computing the total biomass we have:

$$M_i = \sum_{l=1}^{N_i n_i} \frac{H^*}{c} R_i \rightarrow M_i = K R_i, \tag{12}$$

where $K$ is a constant that does not depend on species type. Our assumptions lead to a constraint on the total body mass for species $i$, $M_i = K R_i$, so that a fraction of the energy acquired by individual animals in groups, mediated by $K \propto m_i^0$, gets transformed into body mass $M_i$ at the global population level, i.e. when all individuals of a species are considered. This implies that species with higher biomass will also have higher energy supply rates.

We now insert **Eq. 11** in **Eq. 7** and combine them with the normalisation of frequencies, **Eq. 1**, in order to obtain an analytical expression for the frequency $f_i$ of animal groups of species $i$:

$$M_i = K R_i = m_i \left( \beta R_i m_i^{-3/4} \right) f_i N \rightarrow f_i = \frac{K}{N\beta} m_i^{-1/4} \tag{13}$$

$$\sum_{i=1}^{S} f_i = 1 - f_v \rightarrow \frac{K}{N\beta} \sum_{i=1}^{S} m_i^{-1/4} = 1 - f_v \rightarrow \frac{K}{N\beta} = \frac{1 - f_v}{\sum_{i=1}^{S} m_i^{-1/4}} \rightarrow \tag{14}$$

$$f_i = (1 - f_v) \frac{m_i^{-1/4}}{\sum_{j=1} m_j^{-1/4}}. \tag{15}$$

With our constraint for the total body mass, the frequencies of animal groups scale as a power-law with exponent $-1/4$. This scaling quantity is different from the scaling coefficient $-3/4$ that comes from metabolic theory and which refers to individuals rather than animal groups. Notice that our scaling for animal groups is indeed a consequence of metabolic theory for individuals and hence it assumes validity of allometric scaling in real-world ecosystems. The above approximations allow to express frequencies of mammal groups as a function of the frequency of vector colonies $f_v$.

## Appendix 4

DOI: https://doi.org/10.7554/eLife.32814.017

### Model parameters and initial conditions

The ecomultiplex model adopts a few parameters that are the following:

- Number of total animal groups in the system $N$. We chose $N = 10000$ for getting numerically robust results. The same phenomenology was observed also at smaller scales ($N = 1000$ and $N = 500$).
- Average body mass $m_i$ for individuals of species $i$, which are used for computing the frequencies $f_i$ of animal groups of species $i$ in the ecomultiplex network (see Appendix 3);
- Frequency of vector colonies $f_v$. This quantity determines also the frequency of all the other animal groups in the ecosystem. In the main text we explored a low vector frequency scenario ($f_v = 0.1$) and a high vector frequency scenario ($f_v = 0.25$);
- Average interaction radius $r$: two animal groups are connected in any layer only if they are closer than $r$ in the embedding space. This parameter was tuned numerically to $r = 0.03$ for getting fully connected multiplex networks where on average the parasite could potentially spread across all animal groups in the network;
- Probability $\beta$ for an infected node of transmitting the infection to a susceptible node in the SI model (transmission rate). In order to be compatible with previous results (**Stella et al., 2016**), we chose $\beta = 1$. This choice implies that the parasite spreads exactly accordingly to the topology of the multiplex network, as all contacts among infected and susceptible animal groups lead to parasite transmission with probability $\beta = 1$. Notice that the dependency of parasite spreading on the topology of the ecomultiplex network implies that even if $\beta = 1$ for all species, some species are still more (less) exposed to the parasite, as they can have more (less) connections or be more central along the patterns of infection and thus have more opportunities for getting infected.
- Vectorial layer importance $p_v$, determining the likelihood with which the parasite spreads along a link in the vectorial layer rather than using a link in the food web. When $p_v = 0$ the parasite spreads only across the food web layer. When $p_v = 1$ the parasite spreads only across the vectorial layer. When $p_v = 0.5$, the parasite spreads with equal likelihood across links in the vectorial layer and links in the food web layer. This is a free parameter in the model, ranging between 0 and 1;
- Number of immunised animal groups $\Phi$. We numerically set $\Phi = 1000$ for obtaining statistically significant increases in the global infection time $t^*$ compared to random immunisation within a significance level of 0.05.

All the results reported in the main text are relative to 500 random iterations of the SI model, each one with randomised initial conditions where only 30 adjacent animal groups are infected in the ecomultiplex network. Numerical analysis over the resulting simulations indicated the distributions of infection times being always unimodal, so that there was only a single most frequent value of infection time for every combination of vectorial layer importance and vector frequency. Unimodality justified the use of estimators robust to noise such as medians for computing average quantities for the infection times, as reported in the main text.

Let us also briefly comment on the use of the global infection time as a measure of success of a given strategy. This is related to one important point: independently on the considered strategy, the full multiplex network becomes always infected with the parasite in our simulations (see also the end of 'Model inputs, parameters used and model output' in the main text). This result indicates that the speed of parasite diffusion does not depend on the number of infected hosts in the networked ecosystem but rather only on the time necessary for the parasitic infection to overcome immunised animal groups and flow across the whole network. In other words, time measured in the number of SI updates necessary for reaching the maximum immunisation spread is the only relevant feature for characterising the speed of parasite diffusion. If immunised animal groups of a given species occupy important or central

nodes for the flow of the parasite, then the convergence to the maximum spread state will be slowed down and the time-to-global-infection will be longer. In this way, we use the global infection time as a simple scalar measure for detecting which animal groups occupy special or central nodes in the ecosystem that need to be susceptible for a faster parasite diffusion.

Notice that it is the network connectivity defines which nodes, if susceptible, end up facilitating the flow of the parasite within the multiplex structure. And network connectivity encapsulates the ecological basis of interactions among species in the considered real-world ecosystems. Hence, the global infection time relates to network structure, which in turn is determined by ecological interactions among species communities. In this way, the time-to-maximum infection $t*$ we call global infection time is strongly dependent on the layout of ecological interactions determined through animal diets and parasite prevalence data.

Additional ecological data like species specific transmission rates are actually missing in the relevant literature. With the aim of (i) underlining the importance of ecological interactions, (ii) overcoming ecological data limitations and (iii) following the principle of parameter parsimony, we fixed the infection transmission rates for the SI model to one for all species in the model. In other words, all species have the same probability of catching the parasite when in contact or eating infected species, independently on species type. Let us underline that this choice does not mean that every species is exposed to the parasite with the same probability, since parasite transmission is conditional on the connection to an infected animal group. Again, connectivity is not uniform but rather driven by ecological data, so that also the contact of animal groups of a given species crucially depend on their connectivity. In other words, even if transmission rates are set to 1, their conditionality on connectivity makes some species more exposed to the parasite than others. These different exposure levels to the parasite can make some nodes more central during the parasite flow, as evident from the different increases in infection time when different nodes are immunised.

# Appendix 5

DOI: https://doi.org/10.7554/eLife.32814.018

## Network cartography for ecology

In the main text we used multiplex cartography for visualising topological patterns of different species in our ecomultiplex network model (see *Figure 3* of the main text). The concept of network cartography was introduced by Guimerà and Amaral for distinguishing communities according to their connectivity (*Guimerà and Amaral, 2005*) and it was later generalised by Battiston et al. for multiplex networks (*Battiston et al., 2014*).

A multiplex cartography represents visually, like in a map, the role played by a given node across layers according to its topological features (*Battiston et al., 2014*). An example is reported in Appendix 5 car, where nodes of a fictional 2-layer multiplex network occupy different regions of the cartography. We consider this measure to be of interest for ecological network science because it is a simple yet powerful measure for distinguishing different participation of nodes in multiplex ecological networks.

As in previous works (*Battiston et al., 2014*; *De Domenico et al., 2013*), we consider a cartography based on the following two measures: the multidegree or overlapping degree $K_i$ and the participation coefficient or ratio of uniform link distribution $U_i$ of node $i$. The multidegree $K_i$ is defined as the sum of all the degrees of node $i$ across the $M$ multiplex layers:

$$K_i = \sum_\alpha k_i^{(\alpha)}. \tag{16}$$

where $k_i^{(\alpha)}$ is the degree of node $i$ in layer $\alpha \in \{1, ..., M\}$. The multidegree $K_i$ represents a proxy of the overall local centrality that a node has within the multiplex network. In our ecomultiplex network, the multidegree counts in how many trophic interactions is involved a given animal group and it is therefore a measure accounting for local information (i.e. the neighbourhood of an animal group) but across different interaction types (i.e. all trophic interactions, both eating and contaminative ones).

Differently from (*Battiston et al., 2014*), we consider $K_i$ rather than its standardised counterpart because our ecomultiplex networks do not display Gaussian-like multidegree distributions. We also use multidegree for defining hubs (i.e. animal groups interacting more than the average in the ecosystem) as those nodes being in the 95th percentile of the multidegree distribution.

However, the multidegree does not distinguish between interaction types. Two nodes could have the same multidegree, say $K = 10$, but one could be involved in 10 eating interactions while the other one in five eating interactions and in five contaminative ones, instead. In order to better assess the topology of individual nodes/animal groups in the ecomultiplex network we also consider the ratio $U_i$ of uniform link distribution across layers for node $i$. In formulas, this is defined as:

$$U_i = \frac{M}{M-1} \left[ 1 - \sum_{\alpha=1}^{M} \left( \frac{k_i^{(\alpha)}}{K_i} \right)^2 \right]. \tag{17}$$

$U_i$ ranges between 0 (for nodes that concentrate all their connections in one level only) and 1 (for nodes that distribute connections over all the $M$ layers uniformly). For instance, an animal group with degree five in the food-web and in the vectoriallayer would have a ratio $U$ of uniform link distribution equal to 1. Instead, another animal group with degree 10 in the food-web but 0 in the vectorial layer would have $U = 0$.

As reported also in Appendix 5 car, the couples $(U_i, K_i)$ represent coordinates on a 2D 'map' of a given node: nodes falling in the upper-right part of it are hubs that distribute uniformly their connections across layers while nodes falling in the lower-left part of the map are poorly connected nodes with links mainly in one layer only. Since in both our datasets

vectors contaminate several animal species and are eaten also by several insectivores, we expect for vector groups to fall within the upper-right part of the multiplex cartography.

We report network cartographies for Canastra and Pantanal in the main text (*Figure 3*) and we refer the relative discussion of the cartography results in 'Network analysis' section. Since we simulate ecosystems with $N = 10000$ nodes, we do not visualise individual points but rather clusters of them, obtained by binning the original points in a 2D heat-map. We also plot the average trends of individual species in the ecomultiplex.

**Appendix 5—figure 1.** Scheme on which species are immunised as animal groups in the eco-multiplex network in the different immunisation strategies presented in the main text. The average frequency, serology and hematology of the animal groups immunised in each strategy are presented as well. Error margins indicate standard deviations.
DOI: https://doi.org/10.7554/eLife.32814.019

## Appendix 6

DOI: https://doi.org/10.7554/eLife.32814.020

### Immunisation strategies

*Appendix 6—table 1* summarises all different species considered in each immunisation strategy presented in the main text. Columns represent different immunisation strategies (e.g. All Cricetidae or Parasitised Didelphidae), and rows show species that were immunised (in blue) or susceptible (in red) in these simulations. We also show average frequencies, serologies and hemocultures with error margins (standard deviation) of the animal groups immunised in each strategy. In the main text we show that the best performing strategies are Hemoculture 3 and Parasitised Mammals. Notice that both in Canastra and Pantanal, Hemoculture 3 is not the strategy involving the most abundant species in the ecomultiplex network (cf. the average frequencies).

**Appendix 6—table 1.** Scheme on which species are immunised as animal groups in the ecomultiplex network in the different immunisation strategies presented in the main text.

The average frequency, serology and hematology of the animal groups immunised in each strategy are presented as well. Error margins indicate standard deviations.

Immune ■ and Susceptible ■ Species in the Immunisation Strategies

| CANASTRA Species ↓ | All Cricetidae | All Didelphidae | Large Mammals | Parasitised Mammals | Parasitised Didelphidae | Insectivores | Hemoculture 3 |
|---|---|---|---|---|---|---|---|
| Chrysocyon brachyurus | | | | | | | |
| Leopardus pardalis | | | | | | | |
| Cerdocyon thous | | | | | | | |
| Lycalopex vetulus | | | | | | | |
| Conepatus semistriatus | | | | | | | |
| Didelphis albiventris | | | | | | | |
| Lutreolina crassicaudata | | | | | | | |
| Caluromys philander | | | | | | | |
| Nectomys squamipes | | | | | | | |
| Monodelphis spp | | | | | | | |
| Marmosops incanus | | | | | | | |
| Oxymycterus delator | | | | | | | |
| Cerradomys subflavus | | | | | | | |
| Necromys lasiurus | | | | | | | |
| Akodon montensis | | | | | | | |
| Akodon spp | | | | | | | |
| Gracilinanus agilis | | | | | | | |
| Oligoryzomys spp | | | | | | | |
| Calomysspp | | | | | | | |
| **Mean Frequency (fv = 0.1)** | 640 ± 50 | 480 ± 70 | 200 ± 20 | 560 ± 80 | 520 ± 50 | 510 ± 50 | 400 ± 100 |
| **Mean Frequency (fv = 0.25)** | 540 ± 40 | 400 ± 60 | 170 ± 20 | 470 ± 70 | 430 ± 40 | 430 ± 40 | 400 ± 100 |
| **Mean Serology** | 0.02 ± 0.02 | 0.2 ± 0.1 | 0.3 ± 0.2 | 0.3 ± 0.2 | 0.3 ± 0.3 | 0.12 ± 0.07 | 0.7 ± 0.3 |
| **Mean Hemoculture** | 0.07 ± 0.04 | 0.1 ± 0.1 | 0.2 ± 0.2 | 0.3 ± 0.1 | 0.3 ± 0.2 | 0.07 ± 0.04 | 0.6 ± 0.2 |

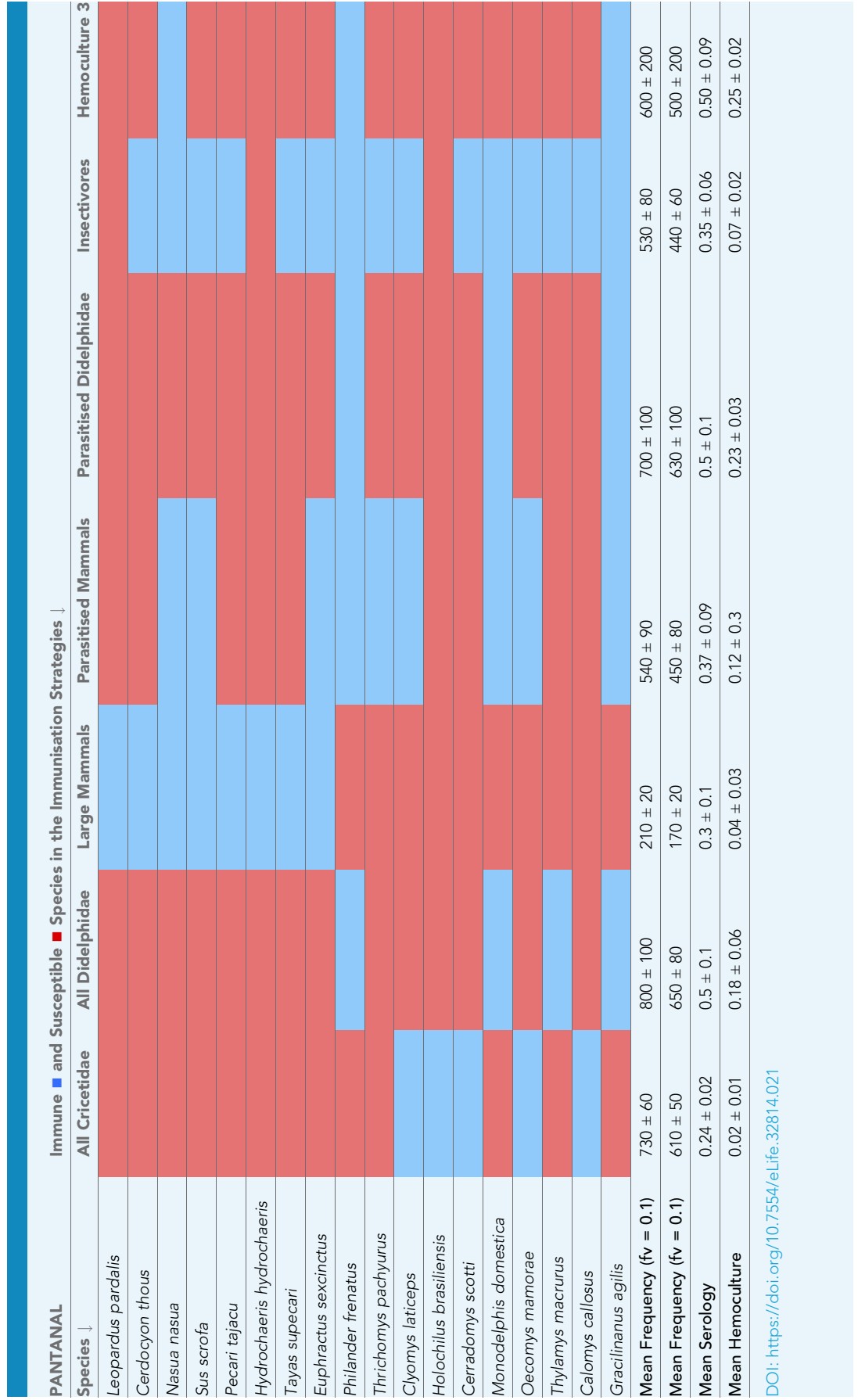

DOI: https://doi.org/10.7554/eLife.32814.021

## Appendix 7

DOI: https://doi.org/10.7554/eLife.32814.022

### Additional Immunisation Strategies

*Appendix 7—figure 1* reports other immunisation strategies we explored, such as:

- Parasitised Cricetidae: immunising the species from the Cricetidae family that are linked to the vector on the vectorial layer;
- Parasitised Prey: immunising the prey species linked to the vector on the vectorial layer;
- Hemoculture (Highest 6): immunising the first six species with the highest likelihood of being found infected with the parasite in field work;
- Serology (Highest 3): immunising the first three species with the highest likelihood of having being found infected with the parasite during their lifetime in field work;
- Serology (Highest 6): immunising the first three species with the highest likelihood of having being found infected with the parasite during their lifetime in field work.

The immunisation strategies reported in *Appendix 7—figure 1* start are compared against the best ecomultiplex strategy from the main text, that is, Parasitised Mammals (in which the species linked to the vector on the vectorial layer are immunised). Interestingly, in terms of infection time increases, Serology (Highest 3) outperforms the ecomultiplex based strategy Parasitised Mammals when vector colonies compose 10% of the ecosystem ($f_v = 0.1$) and when the parasite spreads mainly through the food-web layer (sign test, p-value<$10^{-2}$). In fact, for vectorial layer importance $p_v$>0.5, Serology (Highest 3) and Parasitised Mammals give equivalent results. Notice, however, that Serology (Highest 3) performs consistently below the infection time increase 0.3 when $p_v \leq 0.2$ and $f_v = 0.1$, which is the infection time increase registered for the best epidemiological immunisation strategy, Hemoculture (Highest three in main text). Hence, serology performs worse than another epidemiological immunisation strategy and this is why it was not inserted for discussion in the main text. Notice also that Serology (Highest 6) performs almost equivalently to random immunisation.

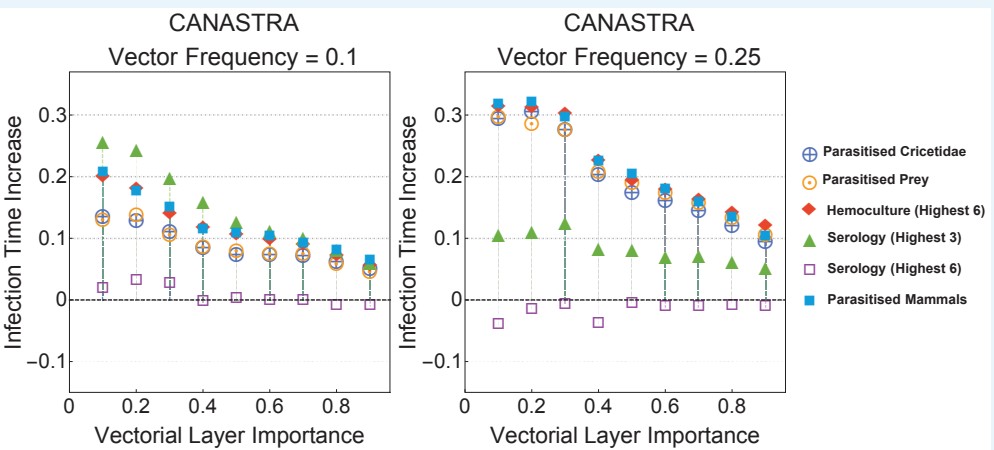

**Appendix 7—figure 1.** Infection time increases in Canastra for immunisation strategies that are not reported in the main text. Visual comparisons are made against the strategy Parasitised Mammals from the main text. For low vector frequency ($f_v = 0.1$) all the reported strategies behave worse than Hemoculture (Highest 3) and were therefore not discussed in the main text.

DOI: https://doi.org/10.7554/eLife.32814.023

Similarly, modifications of the Parasitised Mammals strategy by immunising only specific species according to their biology (e.g. Cricetidae or prey) does not provide any evident improvement in terms of slowing down the parasite spread and increasing the infection time. As a result, Parasitised Cricetidae and Parasitised prey were not inserted in the main text.

Similarly patterns were observed also in Pantanal. Increasing the abundance of vector colonies in the ecomultiplex network leads to worse performances of the serology-based strategies. On the contrary, the measure based on hemoculture provides equivalent result to the immunisation strategies based on the ecomultiplex network structure.

## Appendix 8

DOI: https://doi.org/10.7554/eLife.32814.024

### A null model with equal abundances

In the ecomultiplex model animal groups of different species appear with a power-law frequency, so that animal groups of some species can be considerably more frequent than others. Previous findings indicated how abundance can indeed influence parasite spread among populations, so that a question can naturally arise: are the gaps in performances of immunisation strategies due just to heterogeneity of abundance distribution in the model?

In order to test this research question, we considered a null model equivalent to the scenarios described in the main text but where all species had the same body mass and hence the same group frequency. Results for the Canastra ecosystem are reported in *Appendix 8— figure 1*. The results indicate that even providing equal abundances to different species does not remove the gap in global infection time observed in the main text. Therefore, we conclude that the inequality in abundances of different species cannot fully explain the gaps in immunisation strategies detected, which are rather considered being dependant on the topology of interactions in the ecomultiplex model. In fact, in the null model with equal abundances, animal groups do not differ for their frequency but rather for their topology only.

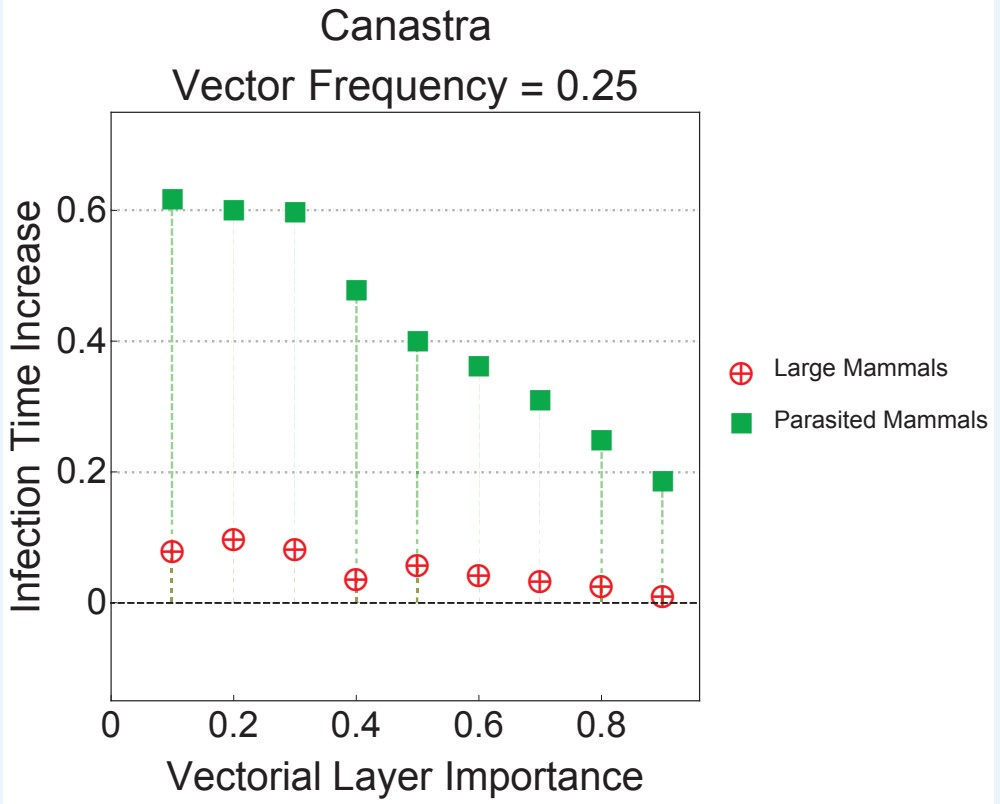

**Appendix 8—figure 1.** The best and the worst performing strategies presented in the main text for the Canastra ecosystem are here presented relatively to the null model with equal abundances. In this null model we consider an ecomultiplex network where all animal groups have equal abundance, that is, occur with equal frequency. The error margins in the plot are the same size of the dots and are based on 500 iterations. Even providing equal abundances to different species does not remove the gap in global infection time that was observed in the main text.

DOI: https://doi.org/10.7554/eLife.32814.025

# Parasite amplification is robust to violations of metabolic theory

The null model with uniform abundance of species (i.e. where all species have the same frequency) allows us to consider how violations of the assumptions of the original model influence the results relative to parasite amplification. In particular, the results presented in the main text rely on the assumption that metabolic theory provides valid scaling relationships for the average number of individuals in animal groups according to body masses. In case this assumption was violated, then no viable analytical approach would allow the estimation of the frequencies of animal groups. In the absence of large-scale ecological data about the distribution and abundance of animals in wildlife ecosystems such as Canastra and Pantanal, one would then be forced to consider rather simple scenarios in which all animal groups have the same frequency.

We explored the ecomultiplex model with uniform species abundance in order to investigate the influence that violations of metabolic theory have on parasite amplification. Results are reported in Appendix 8 equabu2 for the ecological interactions in the Canastra ecosystem. As in the main text, we notice that immunising the three species that are most exposed to the parasite in the wildlife (Hemoculture 3) is still a better immunisation strategy compared to immunising species in contact with the vector on the multiplex structure (Parasitised Mammals). As in the main text, the two strategies display similar performances for increasing values of the vectorial layer importance.

Importantly, even when metabolic theory is violated, parasite amplification is present in the model: immunising the *Leopardus pardalis* leads to substantially higher infection times than immunising another top predator not exposed to the vector. This numerically result confirms that the pattern of parasite amplification observed in the main paper is independent from the specific assumptions relative to metabolic theory and it is rather due to the multiplex structure of ecological interactions.

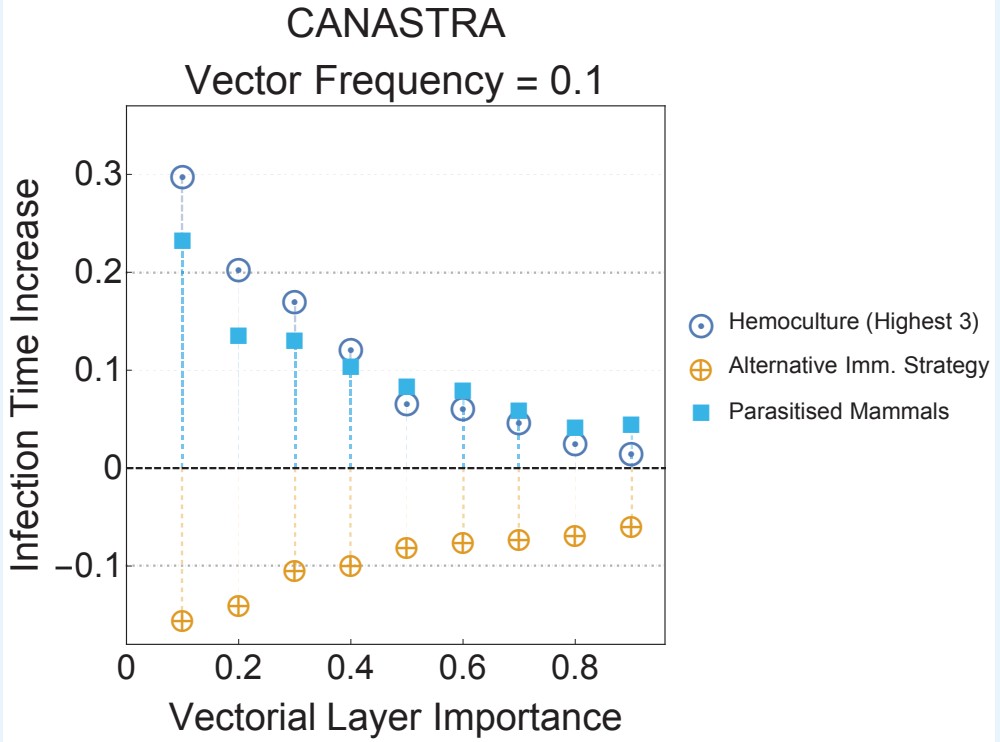

**Appendix 8—figure 2.** The best network-based and the worst performing strategies presented in the main text for the Canastra ecosystem are here presented relatively to the null model with equal abundances. In this null model we consider an ecomultiplex network where all animal groups have equal abundance, that is, occur with equal frequency. The error margins in the

plot are the same size of the dots and are based on 500 iterations. Even providing equal abundances to different species does not remove the gap in global infection time that was observed in the main text.

DOI: https://doi.org/10.7554/eLife.32814.026

