## [Decision Letter]

Thank you for submitting your article "Ecological interactions determine role of species in parasite spread amplification: the ecomultiplex network model" for consideration by *eLife*. Your article has been reviewed by three peer reviewers, and the evaluation has been overseen by a Reviewing Editor and Prabhat Jha as the Senior Editor. The following individuals involved in review of your submission have agreed to reveal their identity: Qianqian Ma (Reviewer #1); Anieke Van Leeuwen (Reviewer #3).

The reviewers have discussed the reviews with one another and the Reviewing Editor has drafted this decision to help you prepare a revised submission.

Summary:

The authors provide an interesting 'ecomultiplex' network modeling framework as a novel means to try to understand and predict responses to control strategies of infectious disease transmission in a complex disease system, that of a multi-host vector borne pathogen. Specifically, they apply their modeling framework to Chagas disease (*Trypanosome cruzi*) transmission, using some data available from two locations in Brazil where sylvatic *T. cruzi* transmission dominates. Specifically, the model is particularly novel in modeling *T. cruzi* systems because it investigates the impacts of trophic relationships (predator-prey) considered with host-parasite relationships on pathogen transmission in the system.

Not surprisingly, reviewers show that vectors play a major role in the sylvatic *T. cruzi* transmission system. The element of 'surprise' in the model, and something which I think many people studying these systems will find interesting, is the model prediction that some predators, often considered to be 'controllers' of infectious disease in a system, can actually facilitate parasite transmission.

Despite its novelty, and the interesting discussion that introducing this model will have among scientists tackling the challenge of trying to understand multi-host pathogen transmission in natural systems, the reviewers raise some substantial concerns in this manuscript (essential revisions) which should be addressed in the revisions.

Essential revisions:

1) Please justify the use of the SI model in this system for the benefit of readers who may not be familiar with the details of *T. cruzi* transmission.

2) Although there is detailed explanation in the Supplementary Information, please provide more information in the Materials and methods explaining model inputs, parameters used, and model outputs, and general methods used so that readers can follow along in the main text better, and improve readers' understanding of the Results.

3) Provide justification for the use of only one vector in the models. Also provide a specific explanation for why you used two different communities for parameter input (section 2.2 and Supplementary Information), what defines their ecology, and in general how they differed.

4) Please provide a more detailed explanation and justification of why you derived species frequencies in the way you did in the Supplementary Information, particularly because all groups have an average interaction radios (section 2.1) and all vectors are grouped together (section 2.2). Basically, section 2 of the Supplementary Information requires improved clarity and improved explanation.

5) Related to #4, The models are constructed based on the assumptions that the metabolic theory provides a good approximation for species and that there is an average size n_i across animal groups of a given species. Please discuss how violations of these assumptions would affect the results. In addition, the home range is simply set to constant across species, which seems unreasonable. Please also discuss how this simple setting and modifying home range can affect the results.

6) In model outputs, the authors only show how the immunization strategies affect infection time t*. Why not present the number of maximum spread, which is also an important indicator of parasite spread?

7) Please explain the assumption and of the use of time-to-infection as measure of strategy success, assuming that the full network will become infected, in particular explain how this assumption relates to the ecological basis in the actual species communities. Is there evidence that the relative rates of processes in the system well established? Please address why, for instance, the transmission rate is set to 1 for all species interactions.

8) Please provide a mention in the Discussion of how in your paper, the model inputs are partly from published data and that the model outputs need to be calibrated and evaluated with additional observed data to truly evaluate model performance.

9) In the Discussion, can you address the parasite amplification phenomenon and how it might relate to or is present in other disease systems in addition to *T. cruzi*?

10) The Supplementary Information section requires increased clarity and English revision.

---

## [Author Response]

Essential revisions:1) Please justify the use of the SI model in this system for the benefit of readers who may not be familiar with the details of T. cruzi transmission.

Thanks for raising this point.

Parasite transmission is modeled as an SI process because the *T. cruzi* infection is lifelong.

In order to provide more clarity and background for those readers that are not familiar with the details of *T. cruzi*, we included more details about *T. cruzi* transmission in the Introduction:

“For instance, *T. cruzi* has a contaminative route of transmission that is mediated by several invertebrate vectors (Triatominae, eng. kissing bug) that gets infected when blood feeding on infected hosts. […] In general sylvatic hosts do not suffer mortality from *T. cruzi* (Kribs-Zaleta, 2010) but the parasite establishes a lifelong infection in almost all of them (Teixeira et al., 2011).”

We also added a sentence referring to the above explanation at the beginning of the Materials and methods section “Susceptible-Infected model on the ecological multiplex network”:

“As explained in the Introduction, we focus on parasites causing lifelong infections in wild hosts. Hence, parasite spread is simulated as a Susceptible-Infected (SI) process (Hastings and Gross, 2012).

2) Although there is detailed explanation in the Supplementary Information, please provide more information in the Materials and methods explaining model inputs, parameters used, and model outputs, and general methods used so that readers can follow along in the main text better, and improve readers' understanding of the Results.

We thank the reviewers for raising this point. We added a subsection at the end of the Materials and methods section highlighting the main inputs, parameters and outputs of the ecomultiplex model, in the main text. We agree this addition makes it easier for the reader to follow along in the main text and understanding of the results.

3) Provide justification for the use of only one vector in the models. Also provide a specific explanation for why you used two different communities for parameter input (section 2.2 and Supplementary Information), what defines their ecology, and in general how they differed.

We thank the reviewer for raising this point. Although several kissing bug species may act as biological vectors for *T. cruzi*, these species function as a single ecological unit since their ecological role is similar. Therefore, previous *T. cruzi* epidemiological models treat several species of kissing bugs as one whole ecological unit (for example in Kribs-Zaleta, 2010). We now provide a justification about the vector compartment and a reference to previous works we got inspiration from in our approach in the new Appendix 2 and in section "Ecological data: Trophic interactions and body masses" of the Materials and methods, where we write:

“Since kissing bugs function as a single ecological unit and previous *T. cruzi* epidemiological models treat the vectors as a single compartment

(Kribs-Zaleta, 2006, 2010), all vector species are grouped as one functional group.”

We also included an extra Appendix 2 related to the Ecology of *T. cruzi* vectors.In this section we present the most common insect vectors described in Canastra and Pantanal regions and their similar habitat preferences.

Ecology of *T. cruzi* vectors

“Approximately 140 species of triatomine bugs have been identified worldwide, however only a few are competent vectors for *T. cruzi* (Lent et al., 1979; Schofield and Galvão, 2009; Rassi Jr et al., 2010). […] Vectors usually prefer to inhabit birds nests on palm trees, trees barks and mammals’ burrows (Gurgel-Gonçalves et al., 2012; Abad-Franch et al., 2009)

Different community structures may affect parasite transmission dynamics. We used data from two communities that differed in species composition and interactions, particularly in the vectorial layer. We included a brief reference to this in the main text, always in section "Ecological data: Trophic interactions and body masses" of the Materials and methods:

“Different community structures may affect parasite transmission dynamics. We used data from two communities that differed in species composition and interactions (Appendix 1).”

We also added the following in the Appendix 1:

“We choose these two datasets because they are the most complete studies describing host species diversity and infection prevalence of *T. cruzi* in natural environments for a long period.”

We also highlighted later at the end of Appendix 1:

“Species interactions differed between places because the species rate of infection also differed. For example, *Leopardus pardalis* interacts with the vector in the vectorial transmission layer in Canastra, but not in Pantanal because of different parasite infection rate data.”

4) Please provide a more detailed explanation and justification of why you derived species frequencies in the way you did in the Supplementary Information, particularly because all groups have an average interaction radios (section 2.1) and all vectors are grouped together (section 2.2). Basically, section 2 of the Supplementary Information requires improved clarity and improved explanation.

We thank the reviewer for this comment. We re-organised Appendix 2 for improved clarity and explanation of our model assumptions and our derivation of the frequencies for animal groups based on the average body mass of individual animals.

We address point 4 together with point 5 as the two points are related.

5) Related to #4, The models are constructed based on the assumptions that the metabolic theory provides a good approximation for species and that there is an average size n_i across animal groups of a given species. Please discuss how violations of these assumptions would affect the results. In addition, the home range is simply set to constant across species, which seems unreasonable. Please also discuss how this simple setting and modifying home range can affect the results.

We thank the reviewer for raising this point. We added Appendix 8 for discussing how violations of metabolic theory (i.e. body mass scaling and the average size n_i across animal groups of a given species) influence our results.

In absence of specific ecological data about abundance of individual species in the considered real-world ecosystems, not considering metabolic theory for allometric scaling of species frequency leads to no prior expectation of the abundance of individual species in an ecosystem. Hence, the simplest reasonable assumption in the above case is to consider uniform abundance of species in the ecosystem. By performing additional numerical simulations, we now show in Appendix 8 that in an ecomultiplex model where all animal species have the same frequencies/abundance, the parasite spread hampering observed in the immunisation strategies using the multiplex structure and the phenomenon of parasite amplification are both still observed, with magnitudes similar to those registered in the model presented in the main text (i.e. the model with metabolic theory).

These numerical experiments represent strong evidence that the results from the ecomultiplex model are robust to violations of assumptions such as allometric scaling from metabolic theory. The robustness of our results to violations of metabolic theory are now briefly discussed in the main text in the Discussion:

“Notice that the above results and the observed mechanism of parasite amplification are robust also to violations of the assumption of metabolic theory as they are present also in null models with animal abundance independent on body mass (Appendix 8).”

Concerning the home ranges, we apologise because the previous explanation of our assumption of equal distances between animal groups was not clear. In the revised Appendix 2 we explain that considering home ranges for computing the biomass of a given species is problematic: home ranges can overlap with each other and animal groups can compete for resources in these overlapping regions, thus ending up exploiting the environment effectively less than what would be possible from the whole home range. In order to consider this reduced energy flow from the environment to animal groups, due to overlaps in home ranges, we introduce the concept of “effective home ranges”, i.e. home ranges having a radius equal to half the average distance between two animal groups of the same species and thus disjointed between each other. As reported in the new Figure 1 in Appendix 2, these “effective” home ranges are smaller than their original counterparts and thus keep into account the reduced energy flow from the environment to animal groups due to overlaps in the original home ranges.

Furthermore, as a consequence of uniform random spatial embedding of animal species over space, the average distance between animal groups of a given species is constant across species. This makes also effective home ranges to be the same across different species and it is ultimately a consequence of our original assumption of uniform random spatial embedding of animal groups in the environment. Although more detailed ecological data about spatial embedding of vector colonies has been recently introduced in the relevant literature (cfr. Gurgel-Goncalves et al., 2017), unfortunately more detailed spatial ecological data about mammals is still unavailable, thus making the uniform random embedding a reasonable first approach in our exploration. We now underlined both in the Appendix 2 and in the main paper that we are considering as constant the effective home ranges of animal groups in this study rather than the biological home ranges for individuals, which we understand can be quite different between predators and prey. When additional data about mobility patterns of animal groups (rather than individuals) becomes available, the effective home ranges might be easily changed within our ecomultiplex formulation.

Let us underline that the assumption of effective home ranges allows to produce analytical results over the frequencies of animal groups but it does not influence the presence and magnitude of results such as parasite amplification or the good performances of the ecomultiplex immunisation protocols.

6) In model outputs, the authors only show how the immunization strategies affect infection time t*. Why not present the number of maximum spread, which is also an important indicator of parasite spread?

We thank the reviewers for raising this point. At the end of the new Materials and methods subsection "Model inputs, parameters used and model outputs" we now report the average total number N_inf_ of infected animal groups when the parasite reaches its maximum spread. We underline that N_inf_ remains statistically compatible with an average value N_inf_ = 8700 \pm 100 across both ecosystems, vectorial probabilities and immunisation strategies. Since in all simulations there can be 10000 – 1000 = 9000 susceptible animal groups (because 1000 are always immunised), then N_inf_ represents 97 \pm 1% of susceptible infected nodes in the ecosystem, which is compatible with 100% when the error bars are considered. This indicates that almost all animal groups are basically always infected or reached by the parasite. Since this number does not depart from its average value at a statistical significance level of 0.05 depending on vectorial layer probabilities or immunisation strategies, in the main text we focus our exposition of the results on the global infection time. We clarified this at the end of the Materials and methods section, where we specify model outputs.

7) Please explain the assumption and of the use of time-to-infection as measure of strategy success, assuming that the full network will become infected, in particular explain how this assumption relates to the ecological basis in the actual species communities. Is there evidence that the relative rates of processes in the system well established? Please address why, for instance, the transmission rate is set to 1 for all species interactions.

We thank the reviewers for raising this point. The use of the global infection time as a measure of success of a given strategy is related to one important point: independently on the considered strategy, the full multiplex network becomes always infected with the parasite in our simulations (see previous point 7). This result indicates that the speed of parasite diffusion does not depend on the number of infected hosts in the networked ecosystem but rather only on the time necessary for the parasitic infection to overcome immunised animal groups and flow across the whole network. In other words, time measured in the number of Susceptible-Infected model updates necessary for reaching the maximum immunisation spread is the only relevant feature for characterising the speed of parasite diffusion. If immunised animal groups of a given species occupy important or central nodes for the flow of the parasite, then the convergence to the maximum spread state will be slowed down and the time-to-global-infection will be longer. In this way, we use the global infection time as a simple scalar measure for detecting which animal groups occupy special or central nodes in the ecosystem that need to be susceptible for a faster parasite diffusion.

Notice that it is the network connectivity defines which nodes, if susceptible, end up facilitating the flow of the parasite within the multiplex structure. And network connectivity encapsulates the ecological basis of interactions among species in the considered real-world ecosystems. Hence, the global infection time relates to network structure, which in turn is determined by ecological interactions among species communities. In this way, the time-to-maximum infection t* we call global infection time is strongly dependent on the layout of ecological interactions determined through animal diets and parasite prevalence data.

Additional ecological data like species specific transmission rates are actually missing in the relevant literature. With the aim of (i) underlining the importance of ecological interactions, (ii) overcoming ecological data limitations and (iii) following the principle of parameter parsimony, we fixed the infection transmission rates for the SI model to 1 for all species in the model. In other words, all species have the same probability of catching the parasite when in contact or eating infected species, independently on species type. Let us underline that this choice does not mean that every species is exposed to the parasite with the same probability, since parasite transmission is conditional on the connection to an infected animal group. Again, connectivity is not uniform but rather driven by ecological data, so that also the contact of animal groups of a given species crucially depend on their connectivity. In other words, even if transmission rates are set to 1, their conditionality on connectivity makes some species more exposed to the parasite than others. These different exposure levels to the parasite can make some nodes more central during the parasite flow, as evident from the different increases in infection time when different nodes are immunised.

We feel this discussion would be highly beneficial for the interested reader and inserted the above paragraphs at the end of Appendix 4.

Furthermore, we now stress this aspect of exposure more in the revised Discussion:

“The exposure to parasite infection in the wildlife is mediated by a network of contacts. […] Further, we avoid arbitrary parameter value definition.”

8) Please provide a mention in the Discussion of how in your paper, the model inputs are partly from published data and that the model outputs need to be calibrated and evaluated with additional observed data to truly evaluate model performance.

We thank the reviewer for the suggestion. In the Discussion we inserted the following mention about further calibration of the model with additional observed data:

“Model inputs for the ecomultiplex model are partly from published data (e.g. animal diets Herrera et al. (2011) and parasite empirical infection rates Herrera et al. (2011); Rocha et al. (2013)). In order to better calibrate and then evaluate model performance, additional observed ecological data should be used. For instance, larger samples for parasite infection rate estimation and, especially, the frequency of animal contacts would both allow for calibration of model parameters such as the SI infection probabilities, which is not feasible with the currently available ecological data.”

9) In the Discussion, can you address the parasite amplification phenomenon and how it might relate to or is present in other disease systems in addition to T. cruzi?

We briefly explain parasite amplification in the main text in the Discussion, where we write that parasite amplification corresponds to "i.e. increased parasite transmission mediated by top predators".

We expect to find predators amplifying transmission in parasites that infect multiple hosts in the food web but also can have different mechanisms of transmission. We made this addiction and gave other examples in the Discussion by also citing relevant works:

“Parasite amplification by predators may also occur in other systems that show multiple transmission routes including trophic transmission, such as in the *Toxoplasma gondii* Dubey (2004) and *Trypanosoma evansi* Herrera et al. (2011) transmission cycles.”

10) The Supplementary Information section requires increased clarity and English revision.

Thanks for the remark, we have double-checked again the appendixes.